# Nanoparticle-Based Dry Powder Inhaler Containing Ciprofloxacin for Enhanced Targeted Antibacterial Therapy

**DOI:** 10.3390/pharmaceutics17040486

**Published:** 2025-04-07

**Authors:** Petra Party, Márk László Klement, Bianca Maria Gaudio, Milena Sorrenti, Rita Ambrus

**Affiliations:** 1Institute of Pharmaceutical Technology and Regulatory Affairs, University of Szeged, 6720 Szeged, Hungary; party.petra@szte.hu (P.P.); klementmark@outlook.com (M.L.K.); 2Department of Drug Sciences, University of Pavia, 27100 Pavia, Italy; bianca.gaudio@gmail.com (B.M.G.); milena.sorrenti@unipv.it (M.S.)

**Keywords:** ciprofloxacin, nanosuspension, dry powder inhaler, wet milling, spray drying, aerodynamic characterization

## Abstract

**Background**: Ciprofloxacin (CIP) is a poorly water-soluble fluoroquinolone-type antibiotic that can be useful in the treatment of lung infections. When the drugs are delivered directly to the lungs, a smaller dosage is needed to achieve the desired effect compared to the oral administration. Moreover, the application of nanoparticles potentially enhances the effectiveness of the treatments while lowering the possible side effects. Therefore, we aimed to develop a “nano-in-micro” structured dry powder inhaler formulation containing CIP. **Methods**: A two-step preparation method was used. Firstly, a nanosuspension was first prepared using a high-performance planetary mill by wet milling. After the addition of different additives (leucine and mannitol), the solid formulations were created by spray drying. The prepared DPI samples were analyzed by using laser diffraction, nanoparticle tracking analysis, scanning electron microscopy, X-ray powder diffraction, and differential scanning calorimetry. The solubility and in vitro dissolution tests in artificial lung fluid and in vitro aerodynamic investigations (Spraytec^®^ device, Andersen Cascade Impactor) were carried out. **Results**: The nanosuspension (D50: 140.0 ± 12.8 nm) was successfully prepared by the particle size reduction method. The DPIs were suitable for inhalation based on the particle diameter and their spherical shape. Improved surface area and amorphization after the preparation processes led to faster drug release. The excipient-containing systems were characterized by large lung deposition (fine particle fraction around 40%) and suitable aerodynamic diameter (between 3 and 4 µm). **Conclusions**: We have successfully formulated a nanosized antibiotic-containing formulation for pulmonary delivery, which could provide a potential treatment for patients with different respiratory infections.

## 1. Introduction

One of the major issues affecting the new drugs under development is the solubility aspect, which affects around 90% of newly developed medications, drugs falling into BCS classes II and IV. There are many approaches to increasing the rate of dissolution, but over the decades, the consistent strategy of interest has been concerned with the reduction in particle size, which can be achieved through nanonization. Drug nanocrystals were introduced in the early 1990s, and the commercialization of the first products started in the early 2000s [1]. Nowadays, the global nanotechnology drug delivery market is at USD 105.95 billion in 2025 and is expected to double by 2034. In this field, the most representative technology segments concern nanoparticles and nanocrystals, and, secondly, liposomes, micelles, and nanotubes, in various branches of medicine such as neurology, oncology, cardiovascular/physiology, anti-inflammatory/immunology, and anti-infective [2].

Nanosuspensions (NS) consist of pure drugs and a minimal proportion of surfactants or polymers to generate a carrier-free colloidal system in a size range between 10 and 1000 nm. The definition of nanomaterials according to the European Union (EU) requires the particle size to be under 100 nm [3]. Pharmaceutical nanoparticles are defined as individual particles with a size below 1 μm. This met the definition of products prepared by nanotechnology according to the U.S. Food and Drug Administration (FDA) [4], which is relevant for bioavailability [5]. Due to their nearly 100% drug load, nanocrystals require fewer excipients, which may be hazardous, and have a higher concentration of active ingredients at the site of action. However, the application of stabilizing agents is indispensable to avoid the aggregation of colloidal particles and thus improve stability. Preparation of NSs is a promising strategy for the efficient delivery of hydrophobic drugs and poorly soluble actives. They have a great potential for the selective and controlled delivery of drugs to target cells and organs [6,7].

For nanocrystal products, oral medication administration is the most popular method. They are available as solid oral dosage forms made from NSs, like tablets, using techniques like lyophilization, or as liquid oral dosage forms, such as suspensions. Another appealing option is parenteral delivery, which requires sterility but has high active loading. Research has also focused on nanocarrier-based active delivery systems for ocular drug delivery, as they are able to overcome many of the biological barriers of the eye and thus improve the ocular bioavailability of drugs used for eye infections and glaucoma. Dermal administration of nanocrystals is also a promising route. The favorable characteristics of these systems can ensure a greater concentration gradient so that a sufficiently large area of direct surface contact of the crystals with the superficial layers of the skin is achieved. Regarding drug delivery to the lungs, drug absorption, and local bioavailability are two aspects that depend on the fraction of drug deposited and dissolved in lung fluids. Mucociliary clearance and drug absorption are two competitive mechanisms that influence drug fate. Since nano-systems are characterized by a high dissolution rate as shown above, they may be particularly efficient when administered by inhalation [8,9]. Due to the large surface area of the lungs, dissolution and penetration are exceptionally fast. Because of their size, nanoparticles can easily enter through the mucus, eliminating the mechanism of mucociliary clearance. The systems could prolong the retention time of inhaled nanoparticles, providing sufficient duration of time for drug release, leading to improved bioavailability. The liberated nanosized drug can effectively reach the epithelium because they are not eliminated by the size-dependent uptake of the alveolar macrophages [10,11,12]. Nanoparticles have advantages in moving through biological barriers and can improve drug uptake into cells through various endocytosis-based pathways as well [13,14,15]. In general, the required dosage can be reduced due to enhanced drug transport [16]. However, the prolonged residence of the particles in the lung may lead to cellular injury, biological responses, and undesired effects. The impacts (e.g., increased reactivity, oxidative stress, cellular injury, and interruption of cellular processes) of nanocrystals are significantly influenced by their properties. Therefore, to characterize nanoparticles for toxicological investigations, a number of nanomaterial characteristics must be considered, including size distribution, surface area, morphology, solubility, chemical composition, and particle agglomeration [17,18,19].

Ciprofloxacin (CIP) is a second-generation fluoroquinolone, patented in 1983 and marketed since 1987 by Bayer, included in the World Health Organization’s list of essential antibiotics. CIP is effective in a wide range of infections, including those that are difficult to treat. Due to its broad-spectrum bactericidal activity, oral efficacy, and good tolerability, it is widely used for the blind therapy of urinary tract infections, acute uncomplicated cystitis, chronic bacterial prostatitis, lower respiratory tract infections, skin infections, typhoid fever, bone (osteochondritis) and joint infections, as well as pyelonephritis. It has been recognized as a second-line agent for the treatment of cholera and tuberculosis in cases of resistance or non-response to first-line anti-tuberculotic drugs. It can be administered orally, intravenously, or through pulmonary delivery for the treatment of chronic Pseudomonas aeruginosa infections in patients with cystic fibrosis [20,21,22,23]. At the physiological pH of the lungs, CIP exists as a zwitterion, with a net neutral charge. The molecule is close to its isoelectric point and, therefore, is very poorly soluble, but it a has slower dissolution rate, providing an extended residence time in the lungs [24,25]. The zwitterionic form has a prolonged terminal elimination half-life when instilled intratracheally as compared with oral delivery [26]. Nanoparticles are often stabilized through electrostatic repulsion. If the active ingredient is neutral, it does not contribute to surface charge, which can increase the likelihood of aggregation. However, adding steric stabilizers or surfactants can improve the stability of the nanoparticles [27].

A potentially interesting field of application of CIP nanocrystals is the preparation of dry powders for pulmonary administration through inhalation. Pulmonary formulations can be liquid preparations that are dispersed in nebulizers or soft mist inhalers. An intermediate between liquid and dry formulation is pressurized metered dose inhaler preparations. Dry powder inhalers (DPIs) are propellant-free systems, so they have lower greenhouse gas emission potential [28]. DPIs are considered the most stable but are the most challenging due to the detrimental characteristics of micronized materials: they come along with high surface energy, which renders them very cohesive, resulting in poor flowability. Thus, DPI formulations need much engineering on the particle level to balance interparticulate forces in the dry powder bulk and ensure sufficient stability during processing and storage, as well as optimal dispersion and fine particle generation during inhalation. In this context, particles with a mean particle diameter between 200 nm and a few micrometers are used because of various aspects in order to obtain the highest dose fraction deposition in the lower airways: they may incorporate and protect the active pharmaceutical ingredient (API) and are suitable for controlled and sustained drug release; and they are considered to overcome biological barriers, such as epi- or endothelial barriers, via various transport mechanisms and are most promising for effective cell uptake. The usage of nanopharmaceuticals for pulmonary delivery allows us to combine the advantages of nanomaterials and the lung as a target [29,30,31].

The emergence of inhaled CIP shows promise for effectively managing chronic lower respiratory tract infections. Inhaled CIP treatment demonstrated superior efficacy in several phases of clinical trials and in preclinical settings [32]. Moreover, the exploration of nanomaterials for respiratory infections is an area of research that is full of potential and has the fastest clinical application [33]. Our research group previously worked with CIP for pulmonary delivery; however, none of the research work was related to nanoparticles [34,35]. In addition, the team worked on nanoparticle-based non-steroidal anti-inflammatory drug-containing formulations [36,37]. In the literature, the following nano-in-micro structured CIP-containing DPIs can be found. There are CIP nanocrystals prepared by anti-solvent precipitation [38], which was followed by spray or freeze drying as solidification methods [39,40,41,42]. Another previously used method is the combination of sonication and lyophilization [43,44]. There is only one wet milling and spray-drying-based CIP-containing article in the literature [45]. We aimed to develop a nanoparticle-based CIP-containing dry powder inhaler for enhanced targeted antibacterial therapy. We applied more efficient wet milling for the preparation of the NS, thereby achieving a high drug loading in the final DPI formulation without the application of an organic solvent. Therefore, the preparation process used and the final inhaled form are also environmentally friendly.

## 2. Materials and Methods

### 2.1. Materials

Ciprofloxacin (CIP) was used as the active pharmaceutical ingredient (Sigma-Aldrich Chemie GmbH, Darmstadt, Germany). Poly-vinyl-alcohol 4–98 (PVA) was used (Sigma Aldrich Chemie GmbH, Darmstadt, Germany), which can promote stable suspension formation due to the steric or electrosteric stabilization of solid particles [46]. The coating effect of PVA prevents nanoparticles from aggregating and ensures a uniform particle size distribution [47]. The application of PVA would minimize the nanoparticle fusions during drying as well as milling. Its decreasing effect on surface tension could result in smaller particles. PVA produces particles with low moisture content, despite its high hydrophilicity [48]. It can be non-toxic when used at the right concentration of PVA, which makes it appropriate for pulmonary applications [36,49]. L-leucine (LEU) (AppliChem GmbH, Darmstadt, Germany) has a low surface energy and forms a hydrophobic shell around drug particles, minimizing cohesion and adhesion between the particles and the attachment to the capsule. This enhances powder flowability and dispersion upon inhalation. Moreover, LEU tends to form wrinkled surfaces when spray-dried, reducing particle density and increasing deep lung deposition by reducing particles [50,51,52]. The application of LEU can lead to moisture protection, therefore improving the product’s physical storage stability [53]. LEU is commonly used in the development of DPIs, and it is well tolerated in pulmonary applications, making it a safe excipient for inhalable formulations [54]. D-mannitol (MAN) (Molar Chemicals Kft, Halásztelek, Hungary) was a matrix former in the spray-dried formulations. It promotes proper, spherical shape and reduces interparticle cohesion, leading to improved aerosol performance. MAN is less hygroscopic than some of the other sugars, e.g., lactose, which is beneficial in terms of better physical and chemical stability of the DPI formulation [55,56]. Additionally, MAN is an FDA-approved excipient for pulmonary delivery and is non-irritant to the lungs [57,58].

### 2.2. Production of the NS by Wet Milling

A high-performance planetary ball mill (Fritsch GmbH Planetary Micro Mill Pulverisette 7, Fritsch, Idar-Oberstein, Germany) was used to reduce the particle size of the CIP. The following material composition was used: 2.00 g of CIP, 18.0 g of 5% (*w*/*w*%) PVA solution, and 20.00 g of ZrO_2_ beads (d = 0.3 mm). The PVA concentration and the ratio of the suspension and the beads were chosen based on a previous work of our team, where a different API and a planetary ball were used [59]. The milling parameters were also optimized previously using factorial design to reduce milling time by increasing the speed of the rotation using the high-performance ball mill. The final milling time was 30 min with a rotational speed of 800 rpm. After milling, the suspension was diluted by adding 180 g of purified water.

### 2.3. DPI Formulations by Spray Drying

Three spray-dried (SPD) compositions were prepared by adding different amounts of LEU and MAN to the NS (Table 1). Inhalable microparticles were produced using a spray dryer equipped with a two-fluid nozzle of 0.7 mm (Büchi Mini Spray Dryer B-191, Büchi, Flawil, Switzerland). The spray-drying properties were as follows: inlet temperature, 130 °C; outlet temperature, 70 °C; aspirator capacity, 75%; airflow rate, 500 l/h; and feed pump rate, 5%. The parameters were based on a previously published work [60] and preliminary experiments, where the particle size, particle size distribution, and the morphology of the dried particles were observed. The properties of the CIP allow the application of high temperatures, which can decrease the humidity in the final product and improve the yield. The applied aspirator capacity also helped the mentioned properties. The final particle size was influenced by the feed rate used as well [61].

### 2.4. Preparation of the Physical Mixtures

Physical mixtures (PM) were prepared from the raw materials to observe the effect of the excipients. The compositions of the PMs were similar to the SPD samples (Table 1). During the experiments, the different qualities of the SPD samples were compared to the PM.

### 2.5. Determination of the Drug Content

The API contents of the different DPIs were determined by dissolving 1.0 mg of powder in 25 mL of methanol and pH 7.4 phosphate buffer (60 + 40 *V*/*V*%). The solutions were filtered (pore size = 0.45 μm and 0.10 μm, Millex-HV filter unit, Millipore Corporation, Bedford, MS, USA) and analyzed by UV/Vis spectrophotometry (ATI-Unicam, Cambridge, UK) at a wavelength of 271 nm. The measurement was carried out three times.

### 2.6. Laser Diffraction-Based Particle Size Measurement

The particle size, particle size distribution (PSD), and specific surface area (SSA) of the NS were determined by laser diffraction (Mastersizer Scirocco 2000, Malvern Instruments Ltd., Worcestershire, UK). The wet dispersion unit was used. The refractive index (RI) of the CIP was adjusted to 1.50. The suspension was measured three times in purified water with stirring at 2000 rpm. Laser diffraction was also applied to determine the particle size, PSD, and SSA of the SPD samples. In this case, the dry dispersion unit was used. The dispersion air pressure was set to 3.0 bar and a vibration feed was applied. Each sample was measured three times. PSD was characterized by the values of D10 (10% of the volume distribution is below this value), D50 (50% of the volume distribution is below this value), and D90 (90% of the volume distribution is below this value) (Equation (1)). The SSA was calculated from the PSD data under the assumption of spherical particles.Span = (D90 − D10)/(D50)(1)

### 2.7. Nanoparticle Tracking Analysis of the NS

The NanoSight NS 3000 device (Malvern Instruments, Worcestershire, UK) for nanoparticle tracking analysis (NTA) was used to obtain high-resolution particle size information about the NS. The instrument was equipped with a 565 nm laser, a high-sensitivity sCMOS camera, and a syringe pump. The CIP suspension was diluted and loaded into the device using a syringe pump speed of 50. The experiment videos were analyzed using NTA 3.4 Build 3.4.4 after being captured in script control mode (3 videos of 30 s per measurement). A total of 1500 frames per sample were examined.

### 2.8. Investigation of the Morphology of the Nanocrystals and the DPIs

The shape of the particles was analyzed using scanning electron microscopy (SEM, Hitachi S4700; Hitachi Ltd., Tokyo, Japan). The investigation conditions were the following: 10 kV high voltage, 10 mA amperage, and 1.3–13.1 mPa air pressure. A high vacuum evaporator and argon atmosphere were applied to make the sputter-coated samples conductive with gold–palladium (Bio-Rad SC 502; VG Microtech, Uckfield, UK). For the implementation of the particle diameter investigation, ImageJ, a public domain image analyzer software, was used (ImageJ 1.53e, https://imagej.net/ij/, accessed on 20 January 2025).

### 2.9. Analysis of the Crystalline Structure

The crystalline structure was investigated using X-ray powder diffraction (XRPD). The Bruker D8 advance diffractometer and the VANTEC-1 detector (Bruker AXS GmbH, Karlsruhe, Germany) were used with Cu KλI radiation. The powders were placed on a flat quartz glass with an etched square. Scanning was performed at a uniform voltage of 40 kV and a current of 40 mA from 3° to 40°, the scanning time constant was 0.1°/min, and the angular step was 0.01°. The DIFFRACplus EVA program was used for the evaluation.

### 2.10. Thermoanalytical Measurement

Differential scanning calorimetry (DSC) measurements were performed with a Mettler Toledo DSC 821e system with the STARe thermal analysis program V9.1 (Mettler Inc., Schwerzenbach, Switzerland). The samples (from 3 to 5 mg) were heated to temperatures between 25 and 300 °C at a rate of 10 °C/min while maintaining a steady flow of argon at a rate of 10 l/h.

### 2.11. Fourier-Transform Infrared Spectroscopy Investigation

The interactions between CIP and the excipients were investigated by Fourier-transform Infrared Spectroscopy (FT-IR), using the AVATAR 330 FT–IR spectrometer (Thermo Nicolet, Thermo Fisher Scientific Inc., Waltham, MA, USA). The samples were homogenized with 150 mg of KBr in an achate mortar, and the mixtures were pressed to prepare pastilles using a Specac^®^ hydraulic press (Specac, Inc., Orpington UK) with a 10-ton pressing force. The spectra were recorded from 4000 to 400 cm^−1^ at an optical resolution of 4 cm−^1^.

### 2.12. Solubility Test

The apparent solubility tests of the SPD products were implemented in 3 mL of artificial lung fluid (0.68 g/L NaCl, 2.27 g/L NaHCO_3_, 0.02 g/L CaCl_2_, 0.1391 g/L NaH_2_PO_4_, 0.37 g/L glycine and 5.56 mL/L 0.1 M H_2_SO_4_) [62]. The pH of the medium was 7.4 ± 0.1. The samples were stirred with a magnetic stirrer at 25 °C for 24 h, filtered (pore size = 0.45 μm, and 0.1 μm, Millex-HV filter unit, Millipore Corporation, Bedford, MS, USA), and then the drug content was analyzed using a UV/VIS spectrophotometer (ATI-Unicam, Cambridge, UK) at a wavelength of 271 nm. The samples were measured in triplicate.

### 2.13. In Vitro Dissolution Test of the DPIs

A modified paddle method (Hanson SR8 Plus, Teledyne Hanson Research, Chatsworth, CA, USA) of the European Pharmacopeia was used to define the release of CIP from the dosage form [63]. At the present time, there are no regulatory requirements for the in vitro dissolution testing of inhaled products. The estimated value of the lung lining fluid is between 10 and 70 mL [64]; therefore, 50 mL of the previously mentioned simulated lung medium was applied. The samples contained 3.25 mg of CIP, which is a tenth of the estimated dose for pulmonary delivery [26]; therefore, the test was carried out in sink conditions. The samples were introduced to the vessel in a powder form. The paddle was rotated at 100 rpm and the measurement was performed up to 60 min at 37 °C. Samples of 5 mL were taken after 5, 10, 15, 30, and 60 min. The medium was replenished in all cases. After filtration (pore size: 0.45 µm, Millex-HV syringe-driven filter unit, Millipore Corporation, Bedford, MA, USA), the dissolved quantity of CIP was determined spectrophotometrically at a wavelength of 271 nm. The measurement was performed three times.

### 2.14. In Vitro Aerodynamic Characterization

The aerosolization properties of the SPD formulations were evaluated in vitro, using an Andersen cascade impactor (ACI, Apparatus D, Copley Scientific Ltd., Nottingham, UK) [65]. The inhalation flow rate was set at 60 L/min (High-capacity Pump model HCP5, Critical Flow Controller model TPK, Copley Scientific Ltd., Nottingham, UK). The actual airflow through the impactor was measured by a mass flow meter (Flow Meter model DFM 2000, Copley Scientific Ltd., Nottingham, UK). The inhalation time was 4 s. The Breezhaler^®^ single-dose device (Novartis International AG, Basel, Switzerland) was applied, which is classified as a low-resistance inhaler. Transparent size 3 Ezeeflo™ hydroxypropyl methylcellulose capsules (ACG—Associated Capsules Pvt. Ltd., Mumbai, India) were filled with the powders. Measurements were carried out in triplicate. To simulate the pulmonary adhesive conditions, the collection plates were coated with a mixture of Span 85 and cyclohexane (1 + 99 *w*/*w*%). After the measurement, the device, capsules, induction port, plates, and filter (A/E glass fiber filter, Pall Corporation, NY, USA) were washed with methanol and pH 7.4 phosphate buffer (60 + 40 *V*/*V*%) to dissolve the deposited amount of MX. The API was quantified by UV/Vis spectrophotometry at a wavelength of 271 nm. Aerodynamic properties were evaluated using Inhalytix^TM^ software (Copley Scientific Ltd., Nottingham, UK). The fine particle fraction (FPF) and mass median aerodynamic diameter (MMAD) values were determined. FPF is defined as the percentage of mass of the particles consisting of API with an MMAD of less than 5 μm divided by the emitted dose of the formulations. The emitted fraction (EF) was also calculated, which is the released fraction from the DPI device. The geometric standard deviation (GSD) was employed to characterize the aerodynamic diameter distribution.

### 2.15. Aerodynamic Particle Size Analysis Using the Spraytec^®^ Device

The aerodynamic diameter was determined using a Spraytec^®^ laser diffractometer equipped with an inhalation cell (Malvern Instruments Ltd., Worcestershire, UK) and ACI. The investigation accounts for the EF of the DPI formulation and measures PSD directly from the inhalation device. SPD formulations were aerosolized from a size 3 HPMC capsule inserted into a Breezhaler^®^ device connected to an induction port of the inhalation cell. The assembly was attached to an ACI, which created a closed system that allowed measurement of the size under controlled circumstances [66]. The inhalation flow rate was set at 60 L/min. The inhalation time was 4 s. Measurements were made in triplicate.

## 3. Results

### 3.1. Characterization of the NS

#### 3.1.1. Results of the Laser Diffraction-Based Particle Size Distribution of the NS

The initial diameter of the API was in the micrometric size range (D50 = 5.98 μm SSA = 1.33 m^2^/g) and the PSD was heterodisperse (Span = 38.70). The particle size of CIP was successfully reduced by wet milling to the nano range (D10 = 120 ± 7.9 nm, D50 = 189.3 ± 11.6 nm, 257.67 ± 14,47 nm) with a Span of 0.727 ± 0.02. PVA coated the CIP particles during milling, which inhibited particle aggregation [47]. SSA significantly increased (34.93 ± 2.23 m^2^/g), which predicts a higher rate of dissolution compared to the raw CIP [67]. The technique can obtain NS in 30 min of milling, increasing the cost-effectiveness of the preparation.

#### 3.1.2. Outcomes of the Nanoparticle Tracking Analysis of the NS

NTA was used to determine the size distribution of the sample using nanoparticle light scattering and Brownian motion. NTA simultaneously detects large and small particles, resulting in a precise particle distribution [68,69]. According to NTA, the D50 of the CIP NS was 140.0 ± 12.8 nm and the PSD was monodisperse (Span = 1.59 ± 0.43) (Figure 1), which met the results of the laser diffraction-based particle size analysis.

#### 3.1.3. Morphology Results of the Dried NS

The prepared NS was dried and investigated by SEM. The solvent was evaporated at 40 °C in a vacuum drying chamber (Binder VD53, BINDER GmbH, Tuttlingen, Germany). In the images (Figure 2), the nanosized API particles were observed on the PVA matrix. The particle size was determined by the ImageJ program and the diameter of the particles (65.60 nm ± 22.70) was smaller in comparison to the results of wet investigation methods.

### 3.2. Characterization of the Dry Powder Inhaler Formulation

#### 3.2.1. Outcomes of the Laser Diffraction-Based Particle Size Measurement

The size of the SPD particles was applicable for pulmonary delivery since the D50 values were in the 1–5 µm range in the case of the nanoCIP_LEU_SPD and nanoCIP_LEU_MAN_SPD (Table 2). The geometric diameter of the LEU- and MAN-free samples was above 5 μm. Moreover, its PSD was heterodisperse (Span > 2.0), given that the higher the Span value, the broader the distribution. PSD was monodisperse in the case of LEU- and MAN-containing products, which is important for targeting the lungs and for the application of standard drug doses [70].

#### 3.2.2. Findings of the Morphology Investigation

On the SEM images (Figure 3) of the DPIs, the effects of the additives were noticeable. The nanoCIP_SPD showed elongated needle-like particles associated with agglomerates. The size of this sample evaluated by ImageJ did not correspond with the findings of the particle size analysis. The average size of the particles was 3.65 ± 1.11 μm for nanoCIP_SPD because the particles tend to be more easily aggregated without LEU and MAN. Based on the picture, the PSD was not monodisperse and the shape of the particles is not preferable. However, a nearly spherical shape was observed in the case of the other two samples, which was the result of the combination of the excipients and the optimized spray-drying method. The spherical form met the requirements of DPIs [71]. Smooth surfaces are not preferred for pulmonary delivery because they tend to increase the interaction between particles, while rough surfaces enhance aerosolization efficiency, which effect was promoted by LEU. Applying MAN, the spherical morphology shifted to a donut-shaped appearance. The internal hollows, reflecting a low density, can lead to improved aerodynamic properties [51]. The diameters of the particles were the following: 3, 4.29 ± 1.14 μm for nanoCIP_LEU_SPD and 4.12 ± 0.95 μm for nanoCIP_LEU_MAN_SPD. The data are the means ± S.D. (*n* = 100 independent measurements), which was similar to the results of the laser diffraction-based measurement. Therefore, particles containing LEU predicted a proper drug delivery to the bronchial and acinar regions as well [72].

#### 3.2.3. Characteristics of the Crystalline Structure

The XRPD was used to characterize the crystalline state of CIP before and after the preparation process in the case of the pulmonary preferable samples. The XRPD pattern of the raw material and the PMs demonstrated the crystalline structure of CIP (Figure 4). In the PMs containing PVA, the three main peaks of CIP are also observable; however, they are of lower intensity at 14.5, 20.5, and 25°. Moreover, between 18 and 20° theta, there is an overlap of the peaks in the sample due to the presence of amorphous PVA in the sample. In the case of the products, the intensities of the characteristic peaks decreased due to the effects of milling and spray drying on the crystallinity of the API. Considering the sample consisting of only CIP and PVA, it can be observed that all the peaks inherent in the crystalline CIP are no longer present in the product. Referring to the nanoCIP_LEU_SPD sample preparation method resulted in a markedly amorphous product: the only peaks that can be distinguished in this case are those of LEU at about 2.5° and the one at 19.5°. A similar conclusion can be drawn for the last sample: the degree of crystallinity of the LEU and MAN also decreased during solidification. The amorphization can lead to faster dissolution of the CIP in comparison to the crystalline drug. Moreover, it can also affect the stability of the final product, which requires further investigation. However, based on the literature, recrystallization does not significantly happen over 1 year in the case of similar “nano-in-micro” systems. In the case of CIP-containing DPIs, the recrystallization did not influence the particle size distribution and aerodynamic performance [28,60,73].

#### 3.2.4. Results of the Thermoanalytical Measurement

The DSC was employed to investigate the CIP melting process in the case of the raw form, PMs, and SPD products (Figure 5). The raw CIP showed a sharp endothermic peak at 271.61 °C, reflecting its melting point and crystallinity. Based on the DSC curves, the amorphization of the CIP is possible. The melting point in the case of CIP_PM indicated the crystallinity of CIP. However, on the curve of CIP_LEU_PM, the starting point around 255 °C detected the melting of LEU not to the CIP. Since there was no observable peak at 260 °C in the case of nanoCIP_SPD, LEU and CIP were solved in the melted MAN. Due to the smaller particle size, a small amount of the remaining CIP crystals in the nanoCIP_SPD melted at a lower temperature than the raw CIP. It is not feasible to determine whether the curves with melting points of 250–260 °C in the case of LEU-containing products relate to CIP or LEU. But, according to the XRPD spectra, the API in the spray-dried samples was in an amorphous state.

#### 3.2.5. Results of the FTIR Analysis

FT-IR spectral analysis was performed to study the possibility of molecular interactions between CIP and the excipients (PVA, MAN, LEU). To identify the changes occurring during nanonization and spray drying, the initial materials, FT-IR spectra of the physical mixtures, and the spray-dried formulations were compared (Figure 6). CIP has characteristic peaks at 1616, 1498, 1285, and 1035 cm^−1^. All the mentioned peaks of CIP were detected in the spray-dried samples, indicating that the preparation process did not cause any structural changes. The peaks related to PVA (3367 and 2943 cm^−1^) were observed in the nanoCIP_PVA, with a shift to 3392 cm^−^¹ due to hydrogen bonding interactions between PVA and CIP. The peaks of LEU (2957, 2871 cm^−1^) did not change during the preparation method. The presence of MAN caused a slight shift in the O-H stretching peak (from 3324 to 3287 cm^−1^), indicating that MAN forms hydrogen bonds with PVA. The findings of the FTIR analysis can affect the dissolution and stability as well.

#### 3.2.6. Effects of the Formulation on the Solubility

The initial solubility of the raw CIP was 0.021 ± 0.002 mg/mL in artificial lung medium. As a result of the increased surface area of CIP, the solubility of the SPD samples improved in both cases: 0.061 ± 0.010 mg/mL of nanoCIP_LEU_SPD, 0.056 ± 0.012 mg/mL of nanoCIP_LEU_MAN_SPD. The reduction in drug particle size in the nano-range led to an increased surface area, which promoted the increase in solubility [74]. The improved solubility exceeds the minimum inhibitory concentration of different pathogens, such as *Pseudomonas aeruginosa*, which is a common cause of bacterial infection in cystic fibrosis [75,76]. Higher concentrations were previously achieved in the sputum of patients with cystic fibrosis, but this could be related to the different composition of the media [77].

#### 3.2.7. Results of the In Vitro Dissolution Test

During the in vitro drug release test, the amount of CIP was the lowest for samples containing raw materials (Figure 7). Approximately 20% of the drug was released from the SPD samples within the first 5 min. These improvements are related to the higher specific surface area, enhanced solubility, and amorphization of the CIP in both cases. Hydrophilic PVA inhibited aggregation and increased polarity, LEU reduced the cohesion between the particles; therefore, a larger amount of CIP was liberated. In all cases, the water-soluble MAN improved the drug dissolution. Additionally, faster drug release could be achieved in the presence of the used excipients due to their combined effect increasing the drug–solvent interactions [78]. The outcomes demonstrated that the nanosized CIP dissolved better than the PMs. Nevertheless, the time was insufficient for the complete drug dose release, which should be taken into consideration when determining the final drug dosage.

#### 3.2.8. Outcomes of the In Vitro Aerodynamic Characterization

The in vitro aerodynamic investigation was carried out by ACI. The distribution of CIP-containing powders on the plates of the stages is shown in Figure 8. It can be concluded that the performance of nanoCIP_LEU_SPD and nanoCIP_LEU_MAN_SPD was quite similar. The application of MAN decreased the deposited amount of powder in the first five stages, which is preferable for lung targeting.

The in vitro aerodynamic results evaluated by the Inhalytix^TM^ software are presented in Table 3. The MMAD results showed the different behavior of the particles during aerosolization. Now, the size of the MAN-containing powder was larger. However, thanks to its hollow structure, the FPF and EF values were more promising in comparison to nanoCIP_LEU_SPD. The FPF values were higher than the commercially available DPI formulations in the Breezhaler^®^ device [79]. Due to the appropriate excipients being employed, the particle was attached neither to the capsule nor the device, and the EF values exceeded 80%. The GSD values were calculated to be less than 2, indicating that the aerodynamic diameter distribution of nanoaggregate microparticles was narrow [80].

#### 3.2.9. Findings of the Aerodynamic Particle Size Analysis Using the Spraytec^®^ Device

According to the Spraytec^®^ test, the D50 values of the samples were between 4.91 and 5.22 µm and showed a narrow PSD (Table 4). The values were similar to the laser diffraction results in Section 3.2.1. The pulmonary applicability in terms of particle size was proved again. However, the increased particle size by MAN was observable, like in the case of the MMAD results of ACI.

## 4. Discussion

Compared to conventional oral and intravenous antibiotic therapies, direct pulmonary delivery offers the advantage of localized drug action, reducing systemic exposure and associated side effects. Additionally, the nanosized drug particles enhance dissolution and absorption in the lungs, which could lead to faster therapeutic onset and improved efficacy, particularly in patients with chronic or resistant bacterial infections.

We have successfully developed nanosized CIP-containing DPI products combining LEU and MAN for the treatment of respiratory infections. To produce the NS, which was only 30 min long, a particle size reduction technique without the use of organic solvents was employed. To combine the advantages of nanoparticles and pulmonary delivery, the NS was solidified by spray drying using LEU and MAN. A comprehensive physicochemical and dosage form investigation was executed. We managed to adjust the particle size of the SPD formulation under 5 μm to target the appropriate regions of the lung. The LEU-containing formulation showed a spherical form, and the LEU- and MAN-containing formulations appeared to have a donut-like form. The dissolution profile of the nanosized CIP-containing samples was improved in comparison to the initial drug, which was promoted by the reduced size and the partial amorphization of the drug. Moreover, the aerodynamic properties also proved to be adequate, the MMAD results were also in the pulmonary adequate range and the FPF values were around 40%. The measurements with the ACI and Spraytec^®^ device clarified that the behavior of the particles changes during aerosolization.

To contextualize our formulation within the existing literature, we compared its performance to previously published formulations. Spray-dried micelles exhibited a slightly higher FPF while maintaining comparable drug dissolution within the first hour [81]. Several studies utilizing spray drying from organic solvent-based nanosuspension technology did not include an aerodynamic performance evaluation [39,82]. Other nanoparticle formulations prepared using organic solvent-based techniques, followed by freeze-drying, demonstrated similar or improved drug release profiles, when aerodynamic investigations were assessed, performance was either slightly inferior or exceeded our findings. Spray-dried formulations containing sugars displayed comparable aerodynamic characteristics [83,84]. Regarding milling techniques, micronized ciprofloxacin (CIP) obtained by spray drying with isopropyl alcohol exhibited a higher FPF but a lower EF [85]. In conclusion, while numerous studies have employed organic solvents in formulation development, our approach offers a safer and more environmentally friendly alternative.

Despite the promising findings, several challenges must be addressed to facilitate the translation of this technology into clinical practice. One major concern is the scalability of the production process. While wet milling and spray drying are well-established techniques, maintaining batch-to-batch consistency in nanoparticle size, crystallinity, and aerodynamic properties during large-scale manufacturing requires further optimization. Process parameters such as milling time, spray-drying conditions, and excipient ratios must be carefully controlled to ensure reproducibility and long-term stability of the formulation. Furthermore, it is important to note the limitations of the work. Our study provides comprehensive in vitro characterization; however, in vivo validation is essential to confirm the efficacy and safety of our formulation. Moreover, the amorphous formulations may present stability challenges, such as recrystallization and moisture absorption; therefore, the stability of the product should be tested. While our work focused on the development and characterization of a novel DPI formulation, direct comparisons with a currently commercially available DPI product could be informative to observe the potential benefits and limitations of our formulation in relation to existing options. Therefore, future directions are the mentioned limitations, such as in vivo animal study in rats to determine the lung concentration of the drug and long-term stability investigation. We plan to conduct comparative tests with different marketed products, ensuring the clinical relevance of the formulations.

## 5. Conclusions

To conclude, the preparation of the “nano-in-micro” structured DPIs containing CIP was carried out using LEU and MAN, offering an environmentally friendly alternative. The formulation can provide the possibility to deliver the antibiotic directly to the lungs, a viable treatment option for respiratory infections, potentially offering a more effective and patient-friendly alternative to existing antibiotic therapies.

## Figures and Tables

**Figure 1 pharmaceutics-17-00486-f001:**
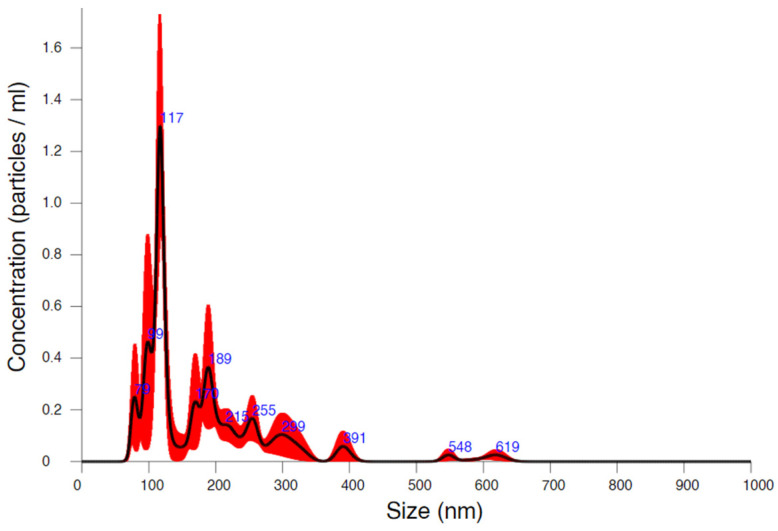
The particle size result of the suspension according to the NTA.

**Figure 2 pharmaceutics-17-00486-f002:**
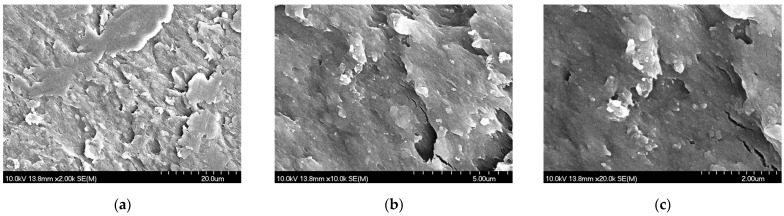
SEM pictures of the dried NS at a magnification of 2000-fold (**a**), 10,000-fold (**b**), and 20,000-fold (**c**).

**Figure 3 pharmaceutics-17-00486-f003:**
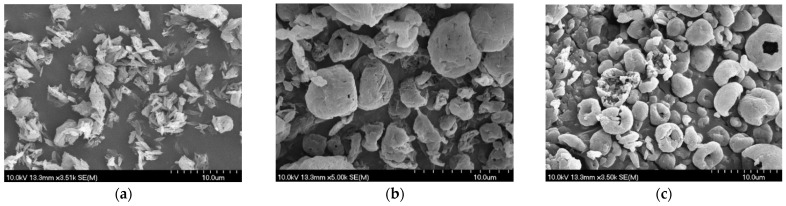
SEM pictures of the SPD samples: (**a**) nanoCIP_SPD; (**b**) nanoCIP_LEU_SPD; (**c**) nanoCIP_LEU_MAN_SPD.

**Figure 4 pharmaceutics-17-00486-f004:**
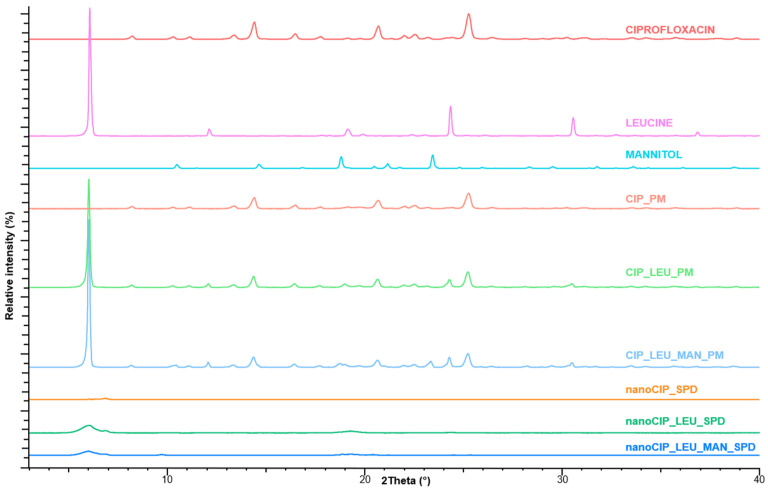
XRPD curves of the initial materials, PMs, and SPD samples.

**Figure 5 pharmaceutics-17-00486-f005:**
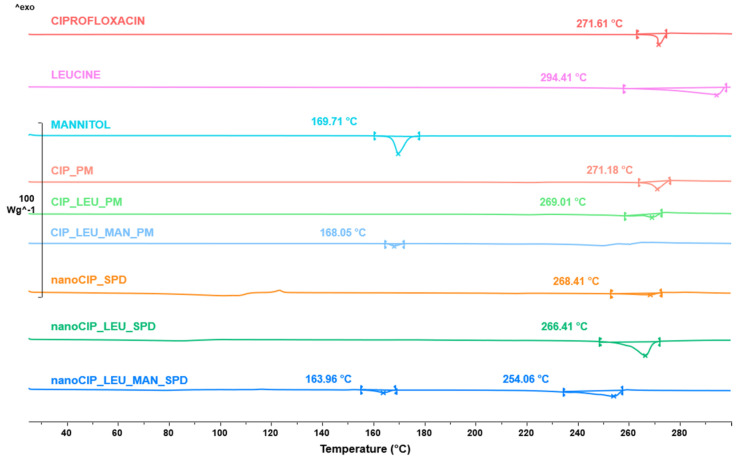
DSC curves of the initial materials, PMs, and SPD samples.

**Figure 6 pharmaceutics-17-00486-f006:**
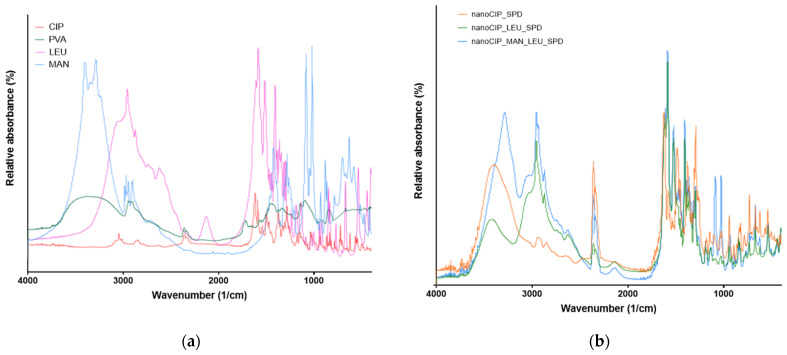
FTIR spectra of the initial materials (**a**) and SPD samples (**b**).

**Figure 7 pharmaceutics-17-00486-f007:**
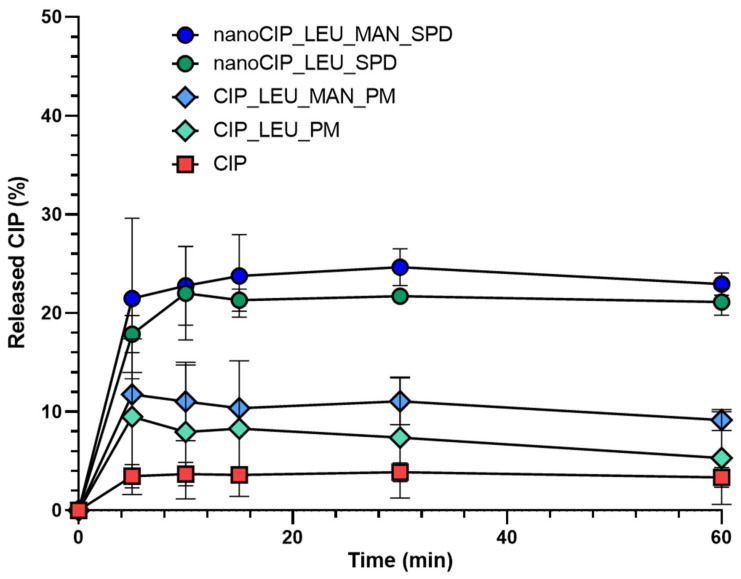
In vitro dissolution results of the DPIs. Data are means ± S.D. (*n* = 3 measurements).

**Figure 8 pharmaceutics-17-00486-f008:**
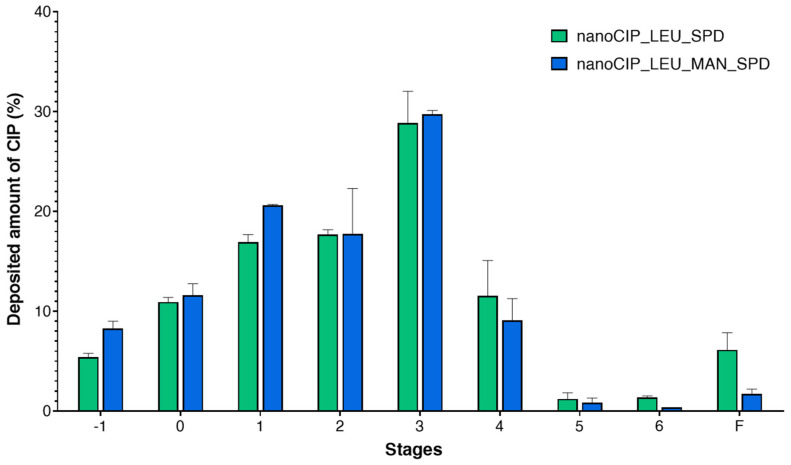
Distribution of the DPIs at a flow rate of 60 l/min. Data are means ± S.D. (*n* = 3 measurements).

**Table 1 pharmaceutics-17-00486-t001:** Composition and API content of the SPD samples and the PMs.

Sample Name	CIP (g)	PVA (g)	LEU (g)	MAN (g)	API Content (%)
nanoCIP_SPD	2.00	0.90	0.00	0.00	66.04 ± 4.37
nanoCIP_LEU_SPD	2.00	0.90	0.40	0.00	40.11 ± 5.85
nanoCIP_LEU_MAN_SPD	2.00	0.90	0.40	0.25	31.94 ± 1.73
CIP_PM	2.00	0.90	0.00	0.00	68.97
CIP_LEU_PM	2.00	0.90	0.40	0.00	60.61
CIP_LEU_MAN_PM	2.00	0.90	0.40	0.25	56.34

**Table 2 pharmaceutics-17-00486-t002:** The particle size, Span, and SSA values of the DPIs. Data are means ± S.D. (*n* = 3 measurements).

Sample Name	D10 (µm)	D50 (µm)	D90 (µm)	Span	SSA (m^2^/g)
nanoCIP_SPD	2.63 ± 0.20	5.66 ± 0.31	15.46 ± 0.50	2.27 ± 0.07	1.23 ± 0.03
nanoCIP_LEU_SPD	2.50 ± 0.05	4.86 ± 0.17	9.41 ± 0.64	1.42 ± 0.08	1.40 ± 0.04
nanoCIP_LEU_MAN_SPD	2.25 ± 0.12	4.59 ± 0.09	9.18 ± 0.44	1.51 ± 0.10	1.52 ± 0.06

**Table 3 pharmaceutics-17-00486-t003:** MMAD, FPF, and EF of the DPIs. Data are means ± S.D. (*n* = 3 independent measurements).

Sample Name	MMAD (µm)	FPF by Size (%)	FPF by Stage (%)	EF (%)	GSD
nanoCIP_LEU_SPD	3.25 ± 0.11	36.49 ± 4.89	41.85 ± 5.96	83.55 ± 0.01	1.99 ± 0.03
nanoCIP_LEU_MAN_SPD	3.71 ± 0.03	41.43 ± 2.88	49.56 ± 3.11	86.65 ± 0.08	1.78 ± 0.08

**Table 4 pharmaceutics-17-00486-t004:** The particle size, Span, and SSA values of the DPIs according to Spraytec^®^ measurement Data are means ± S.D. (*n* = 3 measurements).

Sample Name	D10 (µm)	D50 (µm)	D90 (µm)	Span	SSA (m^2^/g)
nanoCIP_LEU_SPD	2.59 ± 0.05	4.91 ± 0.03	9.06 ± 0.06	1.32 ± 0.03	1.38 ± 0.01
nanoCIP_LEU_MAN_SPD	2.59 ± 0.04	5.22 ± 0.16	10.39 ± 0.56	1.49 ± 0.05	1.32 ± 0.04

## Data Availability

The original contributions presented in the study are included in the article, and further inquiries can be directed to the corresponding author.

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
