# Peer review of "Nanoparticle-Based Dry Powder Inhaler Containing Ciprofloxacin for Enhanced Targeted Antibacterial Therapy"

_pharmaceutics, 2025, doi:10.3390/pharmaceutics17040486_

Round 1
Reviewer 1 Report
Comments and Suggestions for Authors
The manuscript presents a well-structured study on the development of a nanoparticle-based dry powder inhaler (DPI) for ciprofloxacin (CIP) delivery to the lungs. The research is significant in addressing the limitations of conventional pulmonary drug delivery, particularly for bacterial infections. However, there are several areas that require further clarification and critical improvements.
1. Page 1, Line 22-25: The authors claim the DPI showed "large lung deposition (fine particle fraction around 40%)." However, no direct comparison with marketed formulations is provided.
2. Page 2, Line 77-79: The zwitterionic nature of ciprofloxacin is described, but the impact of this property on nanoparticle stability and lung deposition is not addressed.
3. Page 5, Line 128-135: The choice of excipients (PVA, leucine, mannitol) is not adequately justified.
4. Page 5, Line 140-144: The milling parameters (800 rpm, 30 min) are given, but there is no optimization study presented.
5. Page 6, Line 145-151: The spray drying conditions are provided, but no discussion on how these parameters were optimized.
6. Page 9, Line 259-265: The authors report D50 values from laser diffraction but fail to discuss whether the size distribution remains stable over time.
7. Page 12, Line 317-327, Figure 4: XRPD data suggests partial amorphization, but no discussion is provided on how this affects drug dissolution and stability.
8. Page 12, Line 334-344, Figure 5: DSC thermograms show shifts in melting point, indicating possible drug-excipient interactions. Consider performing FTIR analysis to confirm any molecular interactions between CIP and excipients.
9. Page 13, Line 347-352: The solubility of raw CIP (0.021 mg/mL) increased to 0.061 mg/mL in the DPI formulations. However, the practical relevance of this enhancement is not discussed. Compare these values to the solubility required for therapeutic lung concentrations.
10. Page 14, Line 354-364, Figure 6: The dissolution study shows only ~20% drug release in 5 min. It is unclear whether this release rate is sufficient for effective lung targeting.
11. Page 17, Line 397-412: The authors conclude that their formulation is successful but fail to mention potential limitations such as:
The need for in vivo validation.
Potential instability of amorphous formulations over time.
The absence of comparative studies with existing DPI formulations.
12. Page 5, Line 136: "…MAN is less hygroscopic than some of the other sugar,," – Typo: ‘sugar,,’ should be corrected to ‘sugars’
13. Page 17, Line 413: "possibility to delivery antibiotic" – Revise to "possibility to deliver the antibiotic."
14. Page 10, Line 277: "…were observable on the PVA base" – Revise for clarity: "were observed on the PVA matrix."
Comments on the Quality of English Language
The manuscript is generally well-written, but there are occasional grammatical errors, awkward phrasing, and typographical mistakes that affect clarity. A thorough proofreading and minor language polishing are recommended to improve readability and flow.
Author Response
Response to Reviewer 1.
Thank you for reviewing the manuscript. We appreciate your feedback, the modification in the text are highlighted with a green color.
The manuscript presents a well-structured study on the development of a nanoparticle-based dry powder inhaler (DPI) for ciprofloxacin (CIP) delivery to the lungs. The research is significant in addressing the limitations of conventional pulmonary drug delivery, particularly for bacterial infections. However, there are several areas that require further clarification and critical improvements.
- Page 1, Line 22-25: The authors claim the DPI showed "large lung deposition (fine particle fraction around 40%)." However, no direct comparison with marketed formulations is provided.
Thank you for your comment, the result was compared to marketed products in the section 3.2.7.
The FPF values were higher than the commercially available DPI formulations in the Breezhaler® device [1].
- Page 2, Line 77-79: The zwitterionic nature of ciprofloxacin is described, but the impact of this property on nanoparticle stability and lung deposition is not addressed.
Thank you for your comment, the test was modified.
At the physiological pH of the lungs, CIP exists as a zwitterion, with a net neutral charge. The molecule is close to its isoelectric point, therefore very poorly soluble, but it has a slower dissolution rate, providing an extended residence time in the lungs [2,3]. The zwitterionic form has a prolonged terminal elimination half-life, when instilled intratracheally as compared with oral delivery [4]. Nanoparticles are often stabilized through electrostatic repulsion. If the active ingredient is neutral, it does not contribute to surface charge, which can increase the likelihood of aggregation. However, adding steric stabilizers or surfactants can improve the stability of the nanoparticles [5].
- Page 5, Line 128-135: The choice of excipients (PVA, leucine, mannitol) is not adequately justified.
Thank you for your comment, the test was modified.
Poly‐vinyl‐alcohol 4‐98 (PVA) was used (Sigma Aldrich Chemie GmbH, Darmstadt, Germany), which can promote stable suspension formation due to the steric or electrosteric stabilization of solid particles [6]. The coating effect of PVA is preventing nanoparticles from aggregating and ensuring a uniform particle size distribution [7]. The application of PVA would minimize the nanoparticle fusions during drying as well as milling. Its decreasing effect on surface tension could result in smaller particles. PVA produces particles with low moisture content, despite its high hydrophilicity [8]. It can be non-toxic when used at the right concentration of PVA, which makes it appropriate for pulmonary applications [9,10]. L-leucine (LEU) (AppliChem GmbH, Darmstadt, Germany) has a low surface energy and forms a hydrophobic shell around drug particles, minimizing cohesion and adhesion between the particles and the attachment to the capsule. This enhances powder flowability and dispersion upon inhalation. Moreover, LEU tends to form wrinkled surfaces when spray-dried, reducing particle density and increasing deep lung deposition by reducing particle [11–13]. Application of LEU can lead to moisture protection, therefore improving the product’s physical storage stability [14]. LEU is commonly used in the development of DPI-s, it is well tolerated in pulmonary applications, making it a safe excipient for inhalable formulations [15]. D-mannitol (MAN) (Molar Chemicals Kft, Halásztelek, Hungary) was a matrix former in the spray-dried formulations. It promotes proper, spherical shape and reduces interparticle cohesion, leading to improved aerosol performance. MAN is less hygroscopic than some of the other sugars, e.g. lactose, which is beneficial in terms of better physical and chemical stability of the DPI formulation [16,17]. Additionally, MAN is an FDA-approved excipient for pulmonary delivery and is non-irritant to the lungs [18,19].
- Page 5, Line 140-144: The milling parameters (800 rpm, 30 min) are given, but there is no optimization study presented.
The PVA concentration and the ratio of the suspension and the beads were chosen based on a previous work of our team, where a different API and a planetary ball were used [20]. The milling parameters were also optimized previously using factorial design to reduce milling time by increasing the speed of the rotation using the high-performance ball mill.
- Page 6, Line 145-151: The spray drying conditions are provided, but no discussion on how these parameters were optimized.
Thank you for your comment. The text was modified.
The spray drying properties were as follows: inlet temperature: 130 °C, outlet temperature: 70 °C aspirator capacity: 75%, airflow rate: 500 l/h, and feed pump rate: 5%. The parameters were based on a previously published work [21] and preliminary experiments, where the particle size, particle size distribution and the morphology of the dried particles were observed. The properties of the CIP allowed the application of high temperature, which can decrease the humidity in the final product and improve the yield. The applied aspirator capacity also helped the mentioned properties. The final particle size was influenced by the used feed rate as well [22].
- Page 9, Line 259-265: The authors report D50 values from laser diffraction but fail to discuss whether the size distribution remains stable over time.
Thank you for your comment. The text was modified.
The particle size of CIP was successfully reduced by wet milling to the nano range (D10 = 120 ± 7.9 nm, D50 = 189.3 ± 11.6 nm, 257.67 ± 14,47 nm) with a Span of 0.727 ± 0.02.
I attached the particle size distribution curves.
The long-term stability of the prepared suspension has not been tested yet, but we are planning to investigate it in the future. Due to the following spray drying process, long-term stability is not crucial in our case. The suspension particle size distribution stayed constant for five days to perform the solidification. Moreover, based on the literature, a suspension with different API and but with same excipient concentration, and similar suspension and bead ratio, maintained its nano particle size for one month [23].
- Page 12, Line 317-327, Figure 4: XRPD data suggests partial amorphization, but no discussion is provided on how this affects drug dissolution and stability.
Thank you for your comment. The text was modified.
The amorphization can lead to faster dissolution of the CIP in comparison to the crystalline drug. Moreover, it can also affect the stability of the final product, which requires further investigation. However, based on literature, recrystallization does not significantly happen over 1 year in the case of similar “nano-in-micro” systems. In the case of CIP containing DPIs, the recrystallization did not influence the particle size distribution and aerodynamic performance [21,24,25].
- Page 12, Line 334-344, Figure 5: DSC thermograms show shifts in melting point, indicating possible drug-excipient interactions. Consider performing FTIR analysis to confirm any molecular interactions between CIP and excipients.
Thank you for your comment. FTIR analysis was performed. The discussion of DSC investigation was modified.
FT‐IR spectral analysis was performed to study the possibility of molecular interactions between CIP and the excipients (PVA, MAN, LEU). To identify the changes occurring during nanonization and spray-drying. The initial materials, FT‐IR spectra of the physical mixtures and the spray‐dried formulations were compared (Figure 6.). CIP has characteristic peaks at 1616, 1498, 1285 and 1035 cm⁻¹. All the mentioned peaks of CIP were detected in the spray-dried samples, indicating that the preparation process did not cause any structural changes. The peaks related to PVA (3367 and 2943 cm⁻¹) were observed in the nanoCIP_PVA, with a shift to 3392 due to hydrogen bonding interactions between PVA and CIP. The peaks of LEU (2957, 2871 cm-1) did not change during the preparation method. In the presence of MAN, caused a slight shift in the O-H stretching peak (from 3324 to 3287 cm⁻¹), indicating that MAN forms hydrogen bonds with PVA. The findings of the FTIR analysis can affect the dissolution and stability as well.
|
|
|
|
(a) |
(b) |
Figure 6. FTIR spectra of the initial materials (a) and SPD samples (b).
- Page 13, Line 347-352: The solubility of raw CIP (0.021 mg/mL) increased to 0.061 mg/mL in the DPI formulations. However, the practical relevance of this enhancement is not discussed. Compare these values to the solubility required for therapeutic lung concentrations.
The improved solubility exceeds the minimum inhibitory concentration of different pathogens, such as Pseudomonas aeruginosa, which is a common cause of bacterial infection in cystic fibrosis [26,27]. Higher concentrations were previously achieved in sputum of patients with cystic fibrosis, but it could be related to the different composition of the media [28].
- Page 14, Line 354-364, Figure 6: The dissolution study shows only ~20% drug release in 5 min. It is unclear whether this release rate is sufficient for effective lung targeting.
Thank you for your comment. There was a 5-hold increase in dissolution compared to the raw material, which can be more effective in lung targeting. Moreover, the time was insufficient for the complete drugs dose release. We tried to take into consideration the elimination mechanism of the lung such as mucociliary clearance. In literature the time period for the test is usually longer in case of CIP containing formulation [29,30].
- Page 17, Line 397-412: The authors conclude that their formulation is successful but fail to mention potential limitations such as:
- The need for in vivo validation.
- Potential instability of amorphous formulations over time.
- The absence of comparative studies with existing DPI formulations.
- Thank you for your comment. The limitations were mentioned.
Furthermore, it is important to note the limitations of the work. Our study provides comprehensive in vitro characterization; however, in vivo validation is essential to con-firm the efficacy and safety of our formulation. Moreover, the amorphous formulations may present stability challenges, such as recrystallization and moisture absorption, therefore the stability of the product should be tested. While our work focused on the development and characterization of a novel DPI formulation, direct comparisons with a currently commercially available DPI product could be informative to observe the potential benefits and limitations of our formulation in relation to existing options.
- Page 5, Line 136: "…MAN is less hygroscopic than some of the other sugar,," – Typo: ‘sugar,,’ should be corrected to ‘sugars’
Thank you for your comment. It was corrected.
- Page 17, Line 413: "possibility to delivery antibiotic" – Revise to "possibility to deliver the antibiotic."
Thank you for your comment. It was corrected.
- Page 10, Line 277: "…were observable on the PVA base" – Revise for clarity: "were observed on the PVA matrix."
Thank you for your comment. It was corrected.
- Chapman, K.R.; Fogarty, C.M.; Peckitt, C.; Lassen, C. Delivery characteristics and patients ’ handling of two single-dose dry-powder inhalers used in COPD. Int. J. COPD 2011, 6, 353–363.
- Olivera, M.E.; Manzo, R.H.; Junginger, H.E.; Midha, K.K.; Shah, V.P.; Stavchansky, S.; Dressman, J.B. Biowaiver Monographs for Immediate Release Solid Oral Dosage Forms : Ciprofloxacin Hydrochloride. J. Pharm. Sci. 2011, 100, 22–33, doi:10.1002/jps.22259.
- Mcshane, P.J.; Weers, G.; Tarara, T.E.; Haynes, A.; Durbha, P.; Miller, D.P.; Mundry, T.; Operschall, E.; Elborn, J.S. Pulmonary Pharmacology & Therapeutics Cipro fl oxacin Dry Powder for Inhalation ( ciprofloxacin DPI ): Technical design and features of an e ffi cient drug – device combination. Pulm. Pharmacol. Ther. 2018, 50, 72–79, doi:10.1016/j.pupt.2018.03.005.
- Stass, H.; Nagelschmitz, J.; Willmann, S.; Delesen, H.; Gupta, A.; Baumann, S. Inhalation of a Dry Powder Ciprofloxacin Formulation in Healthy Subjects : A Phase I Study. Clin Drug Investig 2013, 419–427, doi:10.1007/s40261-013-0082-0.
- Chin, W.W.L.; Parmentier, J.; Widzinski, M.; Tan, E.N.H.; Gokhale, R. A brief literature and patent review of nanosuspensions to a final drug product. J. Pharm. Sci. 2014, 103, 2980–2999, doi:10.1002/jps.24098.
- Małgorzata, W.; Ostolska, I.; Szewczuk-Karpisz, K.; Chibowski, S.; Terpiłowski, K.; Gun’ko, V.I.; Zarko, V.I. Investigation of the polyvinyl alcohol stabilization mechanism and adsorption properties on the surface. J. Nanoparticle Res. 2015, 17, 1–14, doi:10.1007/s11051-014-2831-2.
- Bartos, C.; Jójárt-Laczkovich, O.; Katona, G.; Budai-Szűcs, M.; Ambrus, R.; Bocsik, A.; Gróf, I.; Deli, M.A.; Szabó-Révész, P. Optimization of a combined wet milling process in order to produce poly(vinyl alcohol) stabilized nanosuspension. Drug Des. Devel. Ther. 2018, 12, 1567–1580, doi:10.2147/DDDT.S159965.
- Wang, Y.; Kho, K.; Cheow, W.S.; Hadinoto, K. A comparison between spray drying and spray freeze drying for dry powder inhaler formulation of drug-loaded lipid-polymer hybrid nanoparticles. Int. J. Pharm. 2012, 424, 98–106, doi:10.1016/j.ijpharm.2011.12.045.
- Chvatal, A.; Alzhrani, R.; Tiwari, A.K.; Ambrus, R.; Szabó-Révész, P.; Boddu, S.H.S. Cytotoxicity of inhalable dry powders in A549 human lung cancer cell line. Farmacia 2018, 66, 172–175.
- Party, P.; Kókai, D.; Burián, K.; Nagy, A.; Hopp, B.; Ambrus, R. Development of extra-fine particles containing nanosized meloxicam for deep pulmonary delivery: in vitro aerodynamic and cell line measurements. Eur. J. Pharm. Sci. 2022, 176, 106247, doi:10.1016/j.ejps.2022.106247.
- Yue-xing, C.; Fei-fei, Y.; Han, W.; Tao-tao, F.; Chun-yu, L.; Li-hui, Q.; Yong-hong, L. The effect of L-leucine on the stabilization and inhalability of spray-dried solid lipid nanoparticles. J. Drug Deliv. Sci. Technol. 2018, 46, 474–481, doi:10.1016/j.jddst.2018.06.011.
- Chvatal, A.; Ambrus, R.; Party, P.; Katona, G.; Jójárt-Laczkovich, O.; Szabó-Révész, P.; Fattal, E.; Tsapis, N. Formulation and comparison of spray dried non-porous and large porous particles containing meloxicam for pulmonary drug delivery. Int. J. Pharm. 2019, 559, 68–75, doi:10.1016/j.ijpharm.2019.01.034.
- Ordoubadi, M.; Shepard, K.B.; Wang, H.; Wang, Z.; Pluntze, A.M.; Churchman, J.P.; Vehring, R. On the Physical Stability of Leucine-Containing Spray-Dried Powders for Respiratory Drug Delivery. Pharmaceutics 2023, 15, 1–26, doi:10.3390/pharmaceutics15020435.
- Sibum, I.; Hagedoorn, P.; Kluitman, M.P.G.; Kloezen, M.; Frijlink, H.W.; Grasmeijer, F. Dispersibility and storage stability optimization of high dose isoniazid dry powder inhalation formulations with L-leucine or trileucine. Pharmaceutics 2020, 12, 1–14, doi:10.3390/pharmaceutics12010024.
- Ke, W.; Yoon, R.; Chang, K.; Chan, H. Engineering the right formulation for enhanced drug delivery. Adv. Drug Deliv. Rev. 2022, 191, 114561, doi:10.1016/j.addr.2022.114561.
- Party, P.; Piszman, Z.I.; Farkas, Á.; Ambrus, R. Comprehensive In Vitro and In Silico Aerodynamic Analysis of High-Dose Ibuprofen- and Mannitol-Containing Dry Powder Inhalers for the Treatment of Cystic Fibrosis. Pharmaceutics 2024, 16, 1–18.
- Hertel, N.; Birk, G.; Scherließ, R. Performance tuning of particle engineered mannitol in dry powder inhalation formulations. Int. J. Pharm. 2020, 586, 119592, doi:10.1016/j.ijpharm.2020.119592.
- Bronchitol Available online: https://www.ema.europa.eu/en/medicines/human/EPAR/bronchitol.
- Shetty, N.; Cipolla, D.; Park, H.; Zhou, Q.T. Physical stability of dry powder inhaler formulations. Expert Opin. Drug Deliv. 2020, 17, 77–96, doi:10.1080/17425247.2020.1702643.
- Bartos, C. Optimization of a combined wet milling process to produce nanosuspension and its transformation into surfactant-free solid compositions to increase the product stability and drug bioavailability, University of Szeged, 2019.
- Benke, E.; Farkas, Á.; Balásházy, I.; Szabó-Révész, P.; Ambrus, R. Stability test of novel combined formulated dry powder inhalation system containing antibiotic: physical characterization and in vitro–in silico lung deposition results. Drug Dev. Ind. Pharm. 2019, 45, 1369–1378, doi:10.1080/03639045.2019.1620268.
- Vehring, R. Pharmaceutical particle engineering via spray drying. Pharm. Res. 2008, 25, 999–1022, doi:10.1007/s11095-007-9475-1.
- Bartos, C.; Motzwickler-Németh, A.; Kovács, D.; Burián, K.; Ambrus, R. Study on the Scale-Up Possibility of a Combined Wet Grinding Technique Intended for Oral Administration of Meloxicam Nanosuspension. Pharmaceutics 2024, 16, 1512.
- Party, P.; Ambrus, R. Investigation of Physico-Chemical Stability and Aerodynamic Properties of Novel “Nano-in-Micro” Structured Dry Powder Inhaler System. Micromachines 2023, 14, 1348, doi:10.3390/mi14071348.
- Karimi, K.; Katona, G.; Csóka, I.; Ambrus, R. Physicochemical stability and aerosolization performance of dry powder inhalation system containing ciprofloxacin hydrochloride. J. Pharm. Biomed. Anal. 2018, 148, 73–79, doi:10.1016/j.jpba.2017.09.019.
- Cirz, R.T.; Neill, B.M.O.; Hammond, J.A.; Head, S.R.; Romesberg, F.E. Defining the Pseudomonas aeruginosa SOS Response and Its Role in the Global Response to the Antibiotic Ciprofloxacin. J. Bacteriol. 2006, 188, 7101–7110, doi:10.1128/JB.00807-06.
- Chalkley, L.J.; Koornhof, H.J. Antimicrobial Activity of Ciprofloxacin against Pseudomonas aeruginosa , Escherichia coli , and Staphylococcus aureus Determined by the Killing Curve Method : Antibiotic Comparisons and Synergistic Interactions. Antimicrob. Agents Chemother. 1985, 28, 331–342.
- Phase, A.I.; Stass, H.; Staab, D. Safety and Pharmacokinetics of Ciprofloxacin Dry Powder for Inhalation in Cystic Fibrosis: A Phase I, Randomized, Single-Dose, Dose-Escalation Study Heino. J. Aerosol Med. Pulm. Drug Deliv. 2014, 27, 1–10, doi:10.1089/jamp.2013.1056.
- Cayli, Y.A.; Sahin, S.; Buttini, F.; Balducci, A.G.; Montanari, S.; Vural, I.; Oner, L.; Akdag, Y.; Sahin, S.; Buttini, F.; et al. Dry powders for the inhalation of ciprofloxacin or levofloxacin combined with a mucolytic agent for cystic fibrosis patients. Drug Dev. Ind. Pharm. 2017, 43, 1378–1389, doi:10.1080/03639045.2017.1318902.
- Liu, Y.; Ma, Y.; Xue, L.; Guan, W.; Wang, Y. Pulmonary multidrug codelivery of curcumin nanosuspensions and ciprofloxacin with N-acetylcysteine for lung infection therapy. J. Drug Deliv. Sci. Technol. 2023, 84, 104474, doi:10.1016/j.jddst.2023.104474.

Reviewer 2 Report
Comments and Suggestions for Authors
This study developed a “nano-in-micro” structured dry power inhaler formulation containing Ciprofloxacin (CIP) for targeted pulmonary delivery. The main findings have demonstrated the successful preparation of NS (D50: 140.0 ± 12.8 nm) by the particle size reduction method and the DPI with an under-5 µm particle diameter and spherical shape. More importantly, the aerodynamic property of the obtained formulation was found to be suitable for pulmonary delivery that MMAD is falling within the optimal range for lung deposition. The FPF achieved was approximately 40%, indicating efficient aerosolization and potential for effective drug delivery to the lower respiratory tract. This approach provides potential treatment for patients with different respiratory infection. Nevertheless, a few aspects of the manuscript could be improved for greater clarity and impact.
1) More details on the selection of excipients (PVA, leucine, and mannitol) should be provided. Why are these specific excipients chosen, and how do they contribute to the stability and performance of the DPI?
2) The stability characterization should be added in this study for demonstrating the formulation potential in practical usage.
3) In Figure 1, why are there so much peaks shown in the graph? Does it indicate the uneven distribution of NS?
4) The discussion is simple-described, so the author’s better to elaborate on the potential clinical implication of these findings. The author should also discuss the scale-up and key challenges of the DPI formulation.
5) The authors should briefly mention future directions for this research, such as in vivo studies or further optimization of the formulation.
6) The manuscript is well-written, but there are a few minor grammatical errors and awkward phrasings that should be corrected. For example, in the abstract, "Faster drug release was the outcome of improved surface area and amorphization that occurred during the preparation methods" could be rephrased for clarity.
Comments on the Quality of English LanguageThe English could be improved to more clearly express the research.
Author Response
Response to Reviewer 2.
Thank you for reviewing the manuscript. We appreciate your feedback, the modification in the text are highlighted with a purple color.
This study developed a “nano-in-micro” structured dry power inhaler formulation containing Ciprofloxacin (CIP) for targeted pulmonary delivery. The main findings have demonstrated the successful preparation of NS (D50: 140.0 ± 12.8 nm) by the particle size reduction method and the DPI with an under-5 µm particle diameter and spherical shape. More importantly, the aerodynamic property of the obtained formulation was found to be suitable for pulmonary delivery that MMAD is falling within the optimal range for lung deposition. The FPF achieved was approximately 40%, indicating efficient aerosolization and potential for effective drug delivery to the lower respiratory tract. This approach provides potential treatment for patients with different respiratory infection. Nevertheless, a few aspects of the manuscript could be improved for greater clarity and impact.
- More details on the selection of excipients (PVA, leucine, and mannitol) should be provided. Why are these specific excipients chosen, and how do they contribute to the stability and performance of the DPI?
Thank you for your comment. The text was modified.
Poly‐vinyl‐alcohol 4‐98 (PVA) was used (Sigma Aldrich Chemie GmbH, Darmstadt, Germany), which can promote stable suspension formation due to the steric or electrosteric stabilization of solid particles [1]. The coating effect of PVA is preventing nanoparticles from aggregating and ensuring a uniform particle size distribution [2]. The application of PVA would minimize the nanoparticle fusions during drying as well as milling. Its decreasing effect on surface tension could result in smaller particles. PVA produces particles with low moisture content, despite its high hydrophilicity [3]. It can be non-toxic when used at the right concentration of PVA, which makes it appropriate for pulmonary applications [4,5]. L-leucine (LEU) (AppliChem GmbH, Darmstadt, Germany) has a low surface energy and forms a hydrophobic shell around drug particles, minimizing cohesion and adhesion between the particles and the attachment to the capsule. This enhances powder flowability and dispersion upon inhalation. Moreover, LEU tends to form wrinkled surfaces when spray-dried, reducing particle density and increasing deep lung deposition by reducing particle [6–8]. Application of LEU can lead to moisture protection, therefore improving the product’s physical storage stability [9]. LEU is commonly used in the development of DPI-s, it is well tolerated in pulmonary applications, making it a safe excipient for inhalable formulations [10]. D-mannitol (MAN) (Molar Chemicals Kft, Halásztelek, Hungary) was a matrix former in the spray-dried formulations. It promotes proper, spherical shape and reduces interparticle cohesion, leading to improved aerosol performance. MAN is less hygroscopic than some of the other sugars, e.g. lactose, which is beneficial in terms of better physical and chemical stability of the DPI formulation [11,12]. Additionally, MAN is an FDA-approved excipient for pulmonary delivery and is non-irritant to the lungs [13,14].
2) The stability characterization should be added in this study for demonstrating the formulation potential in practical usage.
Thank you for your comment. We are planning to conduct stability studies in the future. Based on the stability of PVA and LEU containing “nano-in-micro” structured DPI systems, the stability of our current formulation is promising. The critical attributes of the product can be preserved for one year long [15].
3) In Figure 1, why are there so much peaks shown in the graph? Does it indicate the uneven distribution of NS?
Thank you for your comment. Nanoparticle tracking analysis is a very sensitive method, this is the reason for the numbers of the peaks. But the calculation of the Span value approved, that the particle size distribution is monodisperse. Moreover, on the figure of the laser diffraction-based analysis, only one peak was observable.
4) The discussion is simple-described, so the author’s better to elaborate on the potential clinical implication of these findings. The author should also discuss the scale-up and key challenges of the DPI formulation.
Thank you for your comment. The text was modified.
Compared to conventional oral and intravenous antibiotic therapies, direct pulmonary delivery offers the advantage of localized drug action, reducing systemic exposure and associated side effects. Additionally, the nanosized drug particles enhance dissolution and absorption in the lungs, which could lead to faster therapeutic onset and improved efficacy, particularly in patients with chronic or resistant bacterial infections. In conclusion, it can be said that we have successfully developed nanosized CIP containing DPI products combining LEU and MAN for the treatment of respiratory infections. To produce the NS, which was only 30 minutes long, a particle size reduction technique without the use of organic solvents was employed. To combine the advantages of nanoparticles and pulmonary delivery, the NS was solidified by spray-drying using LEU and MAN. A comprehensive physicochemical and dosage form investigation was executed. We managed to adjust the particle size of the SPD formulation under 5 μm to target the appropriate regions of the lung. The LEU containing formulation showed a spherical form, the LEU and MAN containing formulations appeared to have a donut-like form. The dissolution profile of the nanosized CIP containing samples were improved in comparison to the initial drug, which was promoted by the reduced size and the partial amorphization of the drug. Moreover, the aerodynamic properties also proved to be adequate, the MMAD results were also in the pulmonary adequate range and the FPF values were around 40 %. The measurements with the ACI and Spraytec® device were clarified that the behavior of the particle’s changes during aerosolization. Despite these promising findings, several challenges must be addressed to facilitate the translation of this technology into clinical practice. One major concern is the scalability of the production process. While wet milling and spray-drying are well-established techniques, maintaining batch-to-batch consistency in nanoparticle size, crystallinity, and aerodynamic properties during large-scale manufacturing requires further optimization. Process parameters such as milling time, spray-drying conditions, and excipient ratios must be carefully controlled to ensure reproducibility and long-term stability of the formulation. Furthermore, it is important to note the limitations of the work. Our study provides comprehensive in vitro characterization, however in vivo validation is essential to confirm the efficacy and safety of our formulation. Moreover, the amorphous formulations may present stability challenges, such as recrystallization and moisture absorption, therefore the stability of the product should be tested. While our work focused on the development and characterization of a novel DPI formulation, direct comparisons with a currently commercially available DPI product could be informative to observe the potential benefits and limitations of our formulation in relation to existing options. Therefore, future directions are the mentioned limitations such as in vivo animal study in rats to determine the lung concentration of the drug, long-term stability investigation. We plan to conduct comparative tests with different marketed products, ensuring the clinical relevance of the formulations. To conclude, the preparation of the “nano-in-micro” structured DPIs containing CIP was carried out to provide a possibility to deliver the antibiotic directly to the lungs, a viable treatment option for respiratory infections, potentially offering a more effective and patient-friendly alternative to existing antibiotic therapies.
5) The authors should briefly mention future directions for this research, such as in vivo studies or further optimization of the formulation.
Thank you for your comment. The future directions were added to the text.
Therefore, future directions are the mentioned limitations, such as in vivo animal study in rats to determine the lung concentration of the drug and long-term stability investigation. We plan to conduct comparative tests with different marketed products, ensuring the clinical relevance of the formulations.
6) The manuscript is well-written, but there are a few minor grammatical errors and awkward phrasings that should be corrected. For example, in the abstract, "Faster drug release was the outcome of improved surface area and amorphization that occurred during the preparation methods" could be rephrased for clarity.
Thank you for your comment. The text was modified.
- Małgorzata, W.; Ostolska, I.; Szewczuk-Karpisz, K.; Chibowski, S.; Terpiłowski, K.; Gun’ko, V.I.; Zarko, V.I. Investigation of the polyvinyl alcohol stabilization mechanism and adsorption properties on the surface. J. Nanoparticle Res. 2015, 17, 1–14, doi:10.1007/s11051-014-2831-2.
- Bartos, C.; Jójárt-Laczkovich, O.; Katona, G.; Budai-Szűcs, M.; Ambrus, R.; Bocsik, A.; Gróf, I.; Deli, M.A.; Szabó-Révész, P. Optimization of a combined wet milling process in order to produce poly(vinyl alcohol) stabilized nanosuspension. Drug Des. Devel. Ther. 2018, 12, 1567–1580, doi:10.2147/DDDT.S159965.
- Wang, Y.; Kho, K.; Cheow, W.S.; Hadinoto, K. A comparison between spray drying and spray freeze drying for dry powder inhaler formulation of drug-loaded lipid-polymer hybrid nanoparticles. Int. J. Pharm. 2012, 424, 98–106, doi:10.1016/j.ijpharm.2011.12.045.
- Chvatal, A.; Alzhrani, R.; Tiwari, A.K.; Ambrus, R.; Szabó-Révész, P.; Boddu, S.H.S. Cytotoxicity of inhalable dry powders in A549 human lung cancer cell line. Farmacia 2018, 66, 172–175.
- Party, P.; Kókai, D.; Burián, K.; Nagy, A.; Hopp, B.; Ambrus, R. Development of extra-fine particles containing nanosized meloxicam for deep pulmonary delivery: in vitro aerodynamic and cell line measurements. Eur. J. Pharm. Sci. 2022, 176, 106247, doi:10.1016/j.ejps.2022.106247.
- Yue-xing, C.; Fei-fei, Y.; Han, W.; Tao-tao, F.; Chun-yu, L.; Li-hui, Q.; Yong-hong, L. The effect of L-leucine on the stabilization and inhalability of spray-dried solid lipid nanoparticles. J. Drug Deliv. Sci. Technol. 2018, 46, 474–481, doi:10.1016/j.jddst.2018.06.011.
- Chvatal, A.; Ambrus, R.; Party, P.; Katona, G.; Jójárt-Laczkovich, O.; Szabó-Révész, P.; Fattal, E.; Tsapis, N. Formulation and comparison of spray dried non-porous and large porous particles containing meloxicam for pulmonary drug delivery. Int. J. Pharm. 2019, 559, 68–75, doi:10.1016/j.ijpharm.2019.01.034.
- Ordoubadi, M.; Shepard, K.B.; Wang, H.; Wang, Z.; Pluntze, A.M.; Churchman, J.P.; Vehring, R. On the Physical Stability of Leucine-Containing Spray-Dried Powders for Respiratory Drug Delivery. Pharmaceutics 2023, 15, 1–26, doi:10.3390/pharmaceutics15020435.
- Sibum, I.; Hagedoorn, P.; Kluitman, M.P.G.; Kloezen, M.; Frijlink, H.W.; Grasmeijer, F. Dispersibility and storage stability optimization of high dose isoniazid dry powder inhalation formulations with L-leucine or trileucine. Pharmaceutics 2020, 12, 1–14, doi:10.3390/pharmaceutics12010024.
- Ke, W.; Yoon, R.; Chang, K.; Chan, H. Engineering the right formulation for enhanced drug delivery. Adv. Drug Deliv. Rev. 2022, 191, 114561, doi:10.1016/j.addr.2022.114561.
- Party, P.; Piszman, Z.I.; Farkas, Á.; Ambrus, R. Comprehensive In Vitro and In Silico Aerodynamic Analysis of High-Dose Ibuprofen- and Mannitol-Containing Dry Powder Inhalers for the Treatment of Cystic Fibrosis. Pharmaceutics 2024, 16, 1–18.
- Hertel, N.; Birk, G.; Scherließ, R. Performance tuning of particle engineered mannitol in dry powder inhalation formulations. Int. J. Pharm. 2020, 586, 119592, doi:10.1016/j.ijpharm.2020.119592.
- Bronchitol Available online: https://www.ema.europa.eu/en/medicines/human/EPAR/bronchitol.
- Shetty, N.; Cipolla, D.; Park, H.; Zhou, Q.T. Physical stability of dry powder inhaler formulations. Expert Opin. Drug Deliv. 2020, 17, 77–96, doi:10.1080/17425247.2020.1702643.
- Party, P.; Ambrus, R. Investigation of Physico-Chemical Stability and Aerodynamic Properties of Novel “Nano-in-Micro” Structured Dry Powder Inhaler System. Micromachines 2023, 14, 1348, doi:10.3390/mi14071348.

Reviewer 3 Report
Comments and Suggestions for Authors
The paper Nanoparticle-based dry powder inhaler containing ciprofloxacin for enhanced targeted antibacterial therapy presents the development and characterisation of inhalable microparticles containing ciprofloxacin nanocrystals, which is of interest to the audience of Pharmaceutics. The topic is current and relevant, the research design is sound and justified, and the successful results are clearly presented.
Nevertheless, to further improve the quality of the manuscript, the Authors are encouraged to address the following points:
- The current introduction (esp. lines 55-76) focuses too much on the areas which are not directly relevant to the paper. It would be more beneficial for the reader to give a comprehensive summary of specific advantages and challenges related to nanocrystals in pulmonary delivery, instead of describing other administration routes.
Also, in line 70 there is a fragment about 'low density of nanocrystals on the skin surface' - what does that mean? - Line 46: on what basis do you define the value of 800 nm as the upper size of nanocrystals? It is not in the following ref. 3, it is not based on the official nanotechnology definition (<100 nm) and does not agree with the most common limit in pharmaceutics literature (1000 nm).
- Section 2.2: what size were the zirconia beads?
- In solubility and dissolution testing, the nominal pore size of the filter used (0.45 µm) is bigger than the d50 of the nanosuspension (189 nm), and in section 2.4. there is no mention of filter use. It any case it is very likely that CIP nanoparticles passed through the filter pores. How did you check the efficiency of separation of nanocrystals from solution? How was it ensured that the UV determination of CIP concentration indeed measured only the dissolved compound, and not the remaining undissolved nanocrystals?
- Table 1: It is not explained why the API content is lower in LEU and MAN SPD samples compared to their respective PMs. It would be more adequate to prepare physical mixtures at the composition reflecting the actual drug content in the target systems.
- Section 2.9: More details are needed in the description of the dissolution test: in what form were the samples introduced to the vessel? (Powder? Capsule?) What was the CIP dose? Was the test carried out in sink conditions?
- Section 2.5.3: how was nanosuspension dried as sample preparation for SEM?
- Section 2.6.6.: to avoid misunderstanding, it should be clarified, that the measured solublity values refer to apparent or kinetic solublity (not the thermodynamic value).
- In line 357 it should probably be clarified that the amorphization is partial. It would be interesting to estimate the degree of crystallinity based on the comparison of melting enthalpy values.
- Figure 6: it would be good to change the marker colour or shape to more distinct - the difference between PMs and SPDs might not be visible to all readers.
- The section 5 is titled 'Discussion', but it should be 'Conlusion' (the section is missing). The manuscript however would benefit from expanding the discussion by literature comparison with other research on inhalable CIP nanoparticulate systems. How do the characteristics and performance of the newly developed system compare to those in refs. 23-30, especially where milling or spray drying was used?
- What is the formulation stability? Depending on the degree of CIP amorphization, this would be obviously concerning re. decreased dissolution performance and other nanoparticle changes in time.
- Some references (e.g. 2, 10) are incomplete.
The overall English is good, but some grammar correction is needed. For easier understanding, it would be good to revise the structure of some sentences especially in the introduction (e.g. lines 48-50, 55-59).
Author Response
Response to Reviewer 3.
Thank you for reviewing the manuscript. We appreciate your feedback, the modification in the text are highlighted with orange color.
The paper Nanoparticle-based dry powder inhaler containing ciprofloxacin for enhanced targeted antibacterial therapy presents the development and characterisation of inhalable microparticles containing ciprofloxacin nanocrystals, which is of interest to the audience of Pharmaceutics. The topic is current and relevant, the research design is sound and justified, and the successful results are clearly presented.
Nevertheless, to further improve the quality of the manuscript, the Authors are encouraged to address the following points:
- The current introduction (esp. lines 55-76) focuses too much on the areas which are not directly relevant to the paper. It would be more beneficial for the reader to give a comprehensive summary of specific advantages and challenges related to nanocrystals in pulmonary delivery, instead of describing other administration routes. Also, in line 70 there is a fragment about 'low density of nanocrystals on the skin surface' - what does that mean?
Thank you for your comment. The text was modified.
Regarding drug delivery to the lungs, drug absorption and local bioavailability are two aspects that depend on the fraction of drug deposited and dissolved in lung fluids. Mucociliary clearance and drug absorption are two competitive mechanisms that influence drug fate. Since nano-systems are characterized by a high dissolution rate as shown above, they may be particularly efficient when administered by inhalation [8,9]. Due to the large surface area of the lungs, dissolution and penetration are exceptionally fast. Because of their size, nanoparticles can easily enter through the mucus, eliminating the mechanism of mucociliary clearance. The systems could prolong the retention time of inhaled nanoparticles, providing sufficient duration of time for drug release, leading to improved bioavailability. The liberated nanosized drug can effectively reach the epithelium, because they are not eliminated by the size-dependent uptake of the alveolar macrophages [10–12]. Nanoparticles have advantages in getting through the biological barriers and can improve drug uptake into cells through various endocytosis-based pathways as well [13–15]. In general, the required dosage can be reduced due to enhanced drug transport [16]. However, the prolonged residence of the particles in the lung may lead to cellular injury, biological responses and undesired effects. The impacts (e.g. increased reactivity, oxidative stress, cellular injury and interruption of cellular processes) of nanocrystals are significantly influenced by their properties. Therefore, to characterize nanoparticles for toxicological investigations, a number of nanomaterial characteristics must be considered, including size distribution, surface area, morphology, solubility, chemical composition and particle agglomeration [17–19].
- Line 46: on what basis do you define the value of 800 nm as the upper size of nanocrystals? It is not in the following ref. 3, it is not based on the official nanotechnology definition (<100 nm) and does not agree with the most common limit in pharmaceutics literature (1000 nm).
Thank you for your comment, the test was modified.
Nanosuspensions (NS) consist of pure drugs and a minimal proportion of surfactants or polymers to generate a carrier-free colloidal system in a size range between 10 and 1000 nm. The definition of nanomaterials according to the European Union (EU) requires the particle size to be under 100 nm [3]. Pharmaceutical nanoparticles are defined as individual particles with a size below 1 μm. This met the definition of products prepared by nanotechnology according to the U.S. Food and Drug Administration (FDA) [4]. medium, which is relevant for bioavailability [5]. Due to their nearly 100% drug load, nanocrystals require fewer excipients, which may be hazardous, and have a higher concentration of active ingredients at the site of action.
- Section 2.2: what size were the zirconia beads?
Thank you for your comment. The information was added.
ZrO2 beads (d = 0.3 mm).
- In solubility and dissolution testing, the nominal pore size of the filter used (0.45 µm) is bigger than the d50 of the nanosuspension (189 nm), and in section 2.4. there is no mention of filter use. It any case it is very likely that CIP nanoparticles passed through the filter pores. How did you check the efficiency of separation of nanocrystals from solution? How was it ensured that the UV determination of CIP concentration indeed measured only the dissolved compound, and not the remaining undissolved nanocrystals?
Thank you for your comment. I have repeated the measurement using an extra filter with a smaller pore size (d=0.1 µm) to make sure, that we only detect the solved API. The results did not changed significantly. “the solubility of the SPD samples improved in both cases: 0.061 ± 0.010 mg/ml of nanoCIP_LEU_SPD, 0.056 ± 0.012 mg/ml of nanoCIP_LEU_MAN_SPD.”
In the case of section 2.4. the methanol content ensures that the API is dissolved. There we also filtered the solution to remove the mechanical impurities.
The solutions were filtered (pore size = 0.45 μm, Millex-HV filter unit, Millipore Corporation, Bedford, MS, USA)
- Table 1: It is not explained why the API content is lower in LEU and MAN SPD samples compared to their respective PMs. It would be more adequate to prepare physical mixtures at the composition reflecting the actual drug content in the target systems.
Thank you for your comment. The reason for the lower API content could be related to spray-drying. Some of the API may become attached to the walls of the dryer, cyclone, or collection chamber, leading to losses. Moreover, the nanosized ciprofloxacin particles can be lost in the filter system as well, due to the small size and get carried away with the drying gas. In the future we will prepare physical mixture exactly with the same drug content as the target system.
- Section 2.9: More details are needed in the description of the dissolution test: in what form were the samples introduced to the vessel? (Powder? Capsule?) What was the CIP dose? Was the test carried out in sink conditions?
The samples contained 3.25 mg of CIP, which is a tenth of the estimated dose for pulmonary delivery [26]; therefore, the test was carried out in sink conditions. The samples were introduced to the vessel in a powder form.
Based on the determined solubility of the product was approximately 3 mg of CIP can be solved in 50 ml of lung fluid.
- Section 2.5.3: how was nanosuspension dried as sample preparation for SEM?
Thank you for your comment, the text was modified.
The solvent was evaporated at 40 °C in a vacuum drying chamber (Binder VD53, BINDER GmbH, Tuttlingen, Germany).
- Section 2.6.6.: to avoid misunderstanding, it should be clarified, that the measured solublity values refer to apparentor kinetic solublity (not the thermodynamic value).
Thank you for your comment, the text was modified.
The apparent solubility tests of the SPD products were implemented in 3 ml of artificial lung fluid
- In line 357 it should probably be clarified that the amorphization is partial. It would be interesting to estimate the degree of crystalline based on the comparison of melting enthalpy values.
Thank you for your comment. The XRPD measurement clarified that the API was in an amorphous form in the SPD samples. The discussion section of the DSC results was revised, therefore the crystallinity of CIP was not calculated based on the melting enthalpy values. But in the future we are planning to observe the samples using Hot stage microscopy.
Based on the DSC curves, the amorphization of the CIP is possible. The melting point in the case of CIP_PM indicated the crystallinity of CIP. However, on the curve of CIP_LEU_PM, the starting point around 255 °C detected the melting of LEU not to the CIP. Since there was no observable peak at 260 °C in the case of nanoCIP_SPD, LEU and CIP were solved in the melted MAN. Due to the smaller particle size, a small amount of the remaining CIP crystals in the nanoCIP_SPD melted at a lower temperature than the raw CIP. It is not feasible to determine whether the curves with melting points of 250–260 °C in the case of LEU-containing products relate to CIP or LEU. But according to the XRPD spectra, the API in the LEU containing spray-dried samples was in an amorphous state.
- Figure 6: it would be good to change the marker colour or shape to more distinct - the difference between PMs and SPDs might not be visible to all readers.
Thank you for your comment, the figure was modified.
Figure 7. In vitro dissolution results of the DPIs. Data are means ± S.D. (n = 3 measurements).
- The section 5 is titled 'Discussion', but it should be 'Conlusion' (the section is missing). The manuscript however would benefit from expanding the discussion by literature comparison with other research on inhalable CIP nanoparticulate systems. How do the characteristics and performance of the newly developed system compare to those in refs. 23-30, especially where milling or spray drying was used?
Thank you for your comment. The discussion section was extended, and a conclusion section was added.
To contextualize our formulation within the existing literature, we compared its performance to previously published formulations. Spray-dried micelles exhibited a slightly higher FPF while maintaining comparable drug dissolution within the first hour [16]. Several studies utilizing spray-drying from organic solvent-based nanosuspension technology did not include an aerodynamic performance evaluation [17,18]. Other nanoparticle formulations prepared using organic solvent-based techniques, followed by freeze-drying, demonstrated similar or improved drug release profiles, when aerodynamic investigations were assessed, performance was either slightly inferior or exceeded our findings. Spray-dried formulations containing sugars displayed comparable aerodynamic characteristics [19,20]. Regarding milling techniques, micronized ciprofloxacin (CIP) obtained by spray drying with isopropyl alcohol exhibited a higher fine particle fraction but a lower emitted EF [21]. I n conclusion, while numerous studies have employed organic solvents in formulation development, our approach offers a safer and more environmentally friendly alternative.
To conclude, the preparation of the “nano-in-micro” structured DPIs containing CIP was carried out using LEU and MAN offering an environmentally friendly alternative. The formulation can provide a possibility to deliver the antibiotic directly to the lungs, a viable treatment option for respiratory infections, potentially offering a more effective and patient-friendly alternative to existing antibiotic therapies.
- What is the formulation stability? Depending on the degree of CIP amorphization, this would be obviously concerning re. decreased dissolution performance and other nanoparticle changes in time.
Thank you for your comment. We are planning to conduct stability studies in the future. Based on the stability of PVA and leucine containing “nano-in-micro” structured DPI systems, the stability of our current formulation is promising. The critical attributes of the product can be preserved for one year long [22].
Party, P.; Ambrus, R. Investigation of Physico-Chemical Stability and Aerodynamic Properties of Novel “Nano-in-Micro” Structured Dry Powder Inhaler System. Micromachines 2023, 14, 1348, doi:10.3390/mi14071348.
- Some references (e.g. 2, 10) are incomplete.
Thank you for your comment, the references were revised.
The overall English is good, but some grammar correction is needed. For easier understanding, it would be good to revise the structure of some sentences especially in the introduction (e.g. lines 48-50, 55-59).
Thank you for your comment. The text was modified.
- Malamatari, M.; Taylor, K.M.G.; Malamataris, S.; Douroumis, D.; Kachrimanis, K. Pharmaceutical nanocrystals: production by wet milling and applications. Drug Discov. Today 2018, 23, 534–547, doi:10.1016/j.drudis.2018.01.016.
- Fernández-García, R.; Fraguas-Sánchez, A.I. Nanomedicines for Pulmonary Drug Delivery : Overcoming Barriers in the Treatment of Respiratory Infections and Lung Cancer. Pharmaceutics 2024, 16, 1–33, doi:https://doi.org/10.3390/ pharmaceutics16121584.
- Liu, Q.; Guan, J.; Qin, L.; Zhang, X.; Mao, S. Physicochemical properties affecting the fate of nanoparticles in pulmonary drug delivery. Drug Discov. Today 2020, 25, 150–159, doi:10.1016/j.drudis.2019.09.023.
- Thorley, A.J.; Tetley, T.D. New perspectives in nanomedicine. Pharmacol. Ther. 2013, doi:10.1016/j.pharmthera.2013.06.008.
- Ruge, C.C.; Kirch, J.; Lehr, C.M. Pulmonary drug delivery: From generating aerosols to overcoming biological barriers-therapeutic possibilities and technological challenges. Lancet Respir. Med. 2013, 1, 402–413, doi:10.1016/S2213-2600(13)70072-9.
- Malamatari, M.; Charisi, A.; Malamataris, S.; Kachrimanis, K.; Nikolakakis, I. Spray drying for the preparation of nanoparticle-based drug formulations as dry powders for inhalation. Processes 2020, 8, doi:10.3390/pr8070788.
- Wang, W.; Huang, Z.; Huang, Y.; Zhang, X.; Huang, J.; Cui, Y.; Yue, X.; Ma, C.; Fu, F.; Wang, W.; et al. Pulmonary delivery nanomedicines towards circumventing physiological barriers: Strategies and characterization approaches. Adv. Drug Deliv. Rev. 2022, 185, 114309, doi:10.1016/j.addr.2022.114309.
- Donahue, N.D.; Acar, H.; Wilhelm, S. Concepts of nanoparticle cellular uptake, intracellular trafficking, and kinetics in nanomedicine. Adv. Drug Deliv. Rev. 2019, 143, 68–96, doi:10.1016/j.addr.2019.04.008.
- Torge, A.; Grützmacher, P.; Mücklich, F.; Schneider, M. The influence of mannitol on morphology and disintegration of spray-dried nano-embedded microparticles. Eur. J. Pharm. Sci. 2017, 104, 171–179, doi:10.1016/j.ejps.2017.04.003.
- Bakand, S.; Hayes, A. Toxicological considerations, toxicity assessment, and risk management of inhaled nanoparticles. Int. J. Mol. Sci. 2016, 17, 1–17, doi:10.3390/ijms17060929.
- Hayes, A.J.; Bakand, S. Toxicological perspectives of inhaled therapeutics and nanoparticles. Expert Opin. Drug Metab. Toxicol. 2014, 10, 933–947, doi:10.1517/17425255.2014.916276.
- Kole, E.; Jadhav, K.; Shirsath, N.; Dudhe, P.; Verma, R.K.; Chatterjee, A.; Naik, J. Nanotherapeutics for pulmonary drug delivery: An emerging approach to overcome respiratory diseases. J. Drug Deliv. Sci. Technol. 2023, 81, doi:10.1016/j.jddst.2023.104261.
- Potočnik, J. Comission Recommendation of 18 October 2011 on the definition of nanomaterial. Off. J. Eur. Union 2011, doi:10.7748/ns.24.26.6.s4.
- U.S. Food and Drug Administration Guidance for Industry Considering Whether an FDA-Regulated Product Involves the Application of Nanotechnology Available online: https://www.fda.gov/media/88423/download.
- Stass, H.; Nagelschmitz, J.; Willmann, S.; Delesen, H.; Gupta, A.; Baumann, S. Inhalation of a Dry Powder Ciprofloxacin Formulation in Healthy Subjects : A Phase I Study. Clin Drug Investig 2013, 419–427, doi:10.1007/s40261-013-0082-0.
- Farhangi, M.; Mahboubi, A.; Kobarfard, F.; Vatanara, A.; Mortazavi, S.A. Optimization of a dry powder inhaler of ciprofloxacin-loaded polymeric nanomicelles by spray drying process. Pharm. Dev. Technol. 2019, 24, 584–592, doi:10.1080/10837450.2018.1545237.
- Ma, Y.; Cong, Z.; Wang, Y.; Gao, P. A novel multi-drugs ciprofloxacin-curcumin-N-acetylcysteine co-delivery system based on hybrid nanocrystals for dry powder inhalations. Next Nanotechnol. 2024, 6, 100084, doi:10.1016/j.nxnano.2024.100084.
- Sabuj, M.Z.R.; Rashid, M.A.; Dargaville, T.R.; Islam, N. Stability of Inhaled Ciprofloxacin-Loaded Poly(2-ethyl-2-oxazoline) Nanoparticle Dry Powder Inhaler Formulation in High Stressed Conditions. Pharmaceuticals 2022, 15, 1–16, doi:10.3390/ph15101223.
- Adi, H.; Young, P.M.; Chan, H.K.; Agus, H.; Traini, D. Co-spray-dried mannitol-ciprofloxacin dry powder inhaler formulation for cystic fibrosis and chronic obstructive pulmonary disease. Eur. J. Pharm. Sci. 2010, 40, 239–247, doi:10.1016/j.ejps.2010.03.020.
- Razuc, M.; Piña, J.; Ramírez-rigo, M. V Optimization of Ciprofloxacin Hydrochloride Spray-Dried Microparticles for Pulmonary Delivery Using Design of Experiments. AAPS PharmSciTech 2018, 19, 3085–3096, doi:10.1208/s12249-018-1137-6.
- Cayli, Y.A.; Sahin, S.; Buttini, F.; Balducci, A.G.; Montanari, S.; Vural, I.; Oner, L.; Akdag, Y.; Sahin, S.; Buttini, F.; et al. Dry powders for the inhalation of ciprofloxacin or levofloxacin combined with a mucolytic agent for cystic fibrosis patients. Drug Dev. Ind. Pharm. 2017, 43, 1378–1389, doi:10.1080/03639045.2017.1318902.
- Party, P.; Ambrus, R. Investigation of Physico-Chemical Stability and Aerodynamic Properties of Novel “Nano-in-Micro” Structured Dry Powder Inhaler System. Micromachines 2023, 14, 1348, doi:10.3390/mi14071348.

Round 2
Reviewer 1 Report
Comments and Suggestions for Authors
The revisions have significantly improved the clarity, scientific rigor, and overall quality of the manuscript. The responses provided are satisfactory, and the necessary modifications have been incorporated appropriately appreciate the authors' efforts in refining the manuscript, and I find it now suitable for publication.
Author Response
Dear Reviewer,
thanks for your comments.
Reviewer 3 Report
Comments and Suggestions for Authors
The literature citation is missing for the fragment in lines 385-388.
Author Response
Dear reviewer,
thanks for your remark, it was corrected.